# Light-driven synchronization of optogenetic clocks

Maria Cristina Cannarsa[1,2], Filippo Liguori[1,3], Nicola Pellicciotta[1,4], Giacomo Frangipane[1,4]*, Roberto Di Leonardo[1,4]*

[1]Department of Physics, Sapienza University of Rome, Roma, Italy; [2]Department of Biology and Biotechnology Charles Darwin, Sapienza University of Rome, Rome, Italy; [3]Center for Life Nano & Neuro Science, Fondazione Istituto Italiano di Tecnologia (IIT), Roma, Italy; [4]NANOTEC-CNR, Soft and Living Matter Laboratory, Institute of Nanotechnology, Rome, Italy

**\*For correspondence:**
giacomo.frangipane@uniroma1.it (GF);
roberto.dileonardo@uniroma1.it (RDL)

**Competing interest:** The authors declare that no competing interests exist.

**Abstract** Synthetic genetic oscillators can serve as internal clocks within engineered cells to program periodic expression. However, cell-to-cell variability introduces a dispersion in the characteristics of these clocks that drives the population to complete desynchronization. Here, we introduce the optorepressilator, an optically controllable genetic clock that combines the repressilator, a three-node synthetic network in *E. coli*, with an optogenetic module enabling to reset, delay, or advance its phase using optical inputs. We demonstrate that a population of optorepressilators can be synchronized by transient green light exposure or entrained to oscillate indefinitely by a train of short pulses, through a mechanism reminiscent of natural circadian clocks. Furthermore, we investigate the system's response to detuned external stimuli observing multiple regimes of global synchronization. Integrating experiments and mathematical modeling, we show that the entrainment mechanism is robust and can be understood quantitatively from single cell to population level.

## eLife assessment

This study presents a light-entrainable synthetic oscillator in bacteria, the optorepressilator. The authors develop a toolbox using optogenetics that makes the cellular oscillator easily controllable. This toolbox is **valuable**, contributing both to bioengineering and to the understanding of biological dynamical systems. The comparison with a mathematical model, population, and single-cell measurements demonstrate **convincingly** that the planned system was achieved and is suitable to control and study biological oscillators.

## Introduction

Genetic clocks keep time within living organisms in order to program periodic behaviors like circadian rhythms. An intensive genetic analysis of the complex landscape of these biological oscillations has revealed that, from bacteria and fungi to plants and animals, these clocks share similar motifs in the underlying gene regulatory networks (*Young and Kay, 2001*; *Cookson et al., 2009*). From a reverse perspective, synthetic biologists have tackled the problem of designing from scratch minimal gene networks that can produce periodic patterns of gene expression (*Elowitz and Leibler, 2000*; *Atkinson et al., 2003*; *Stricker et al., 2008*; *Purcell et al., 2010*). Very early numerical simulations had shown that networks with an odd number of cyclically connected genes exhibit robust oscillations (*Fraser and Tiwari, 1974*). The repressilator was the first experimental realization of a synthetic genetic clock based on this principle (*Elowitz and Leibler, 2000*). As shown schematically in *Figure 1*, the repressilator is a minimal network consisting of three transcription factors mutually repressing

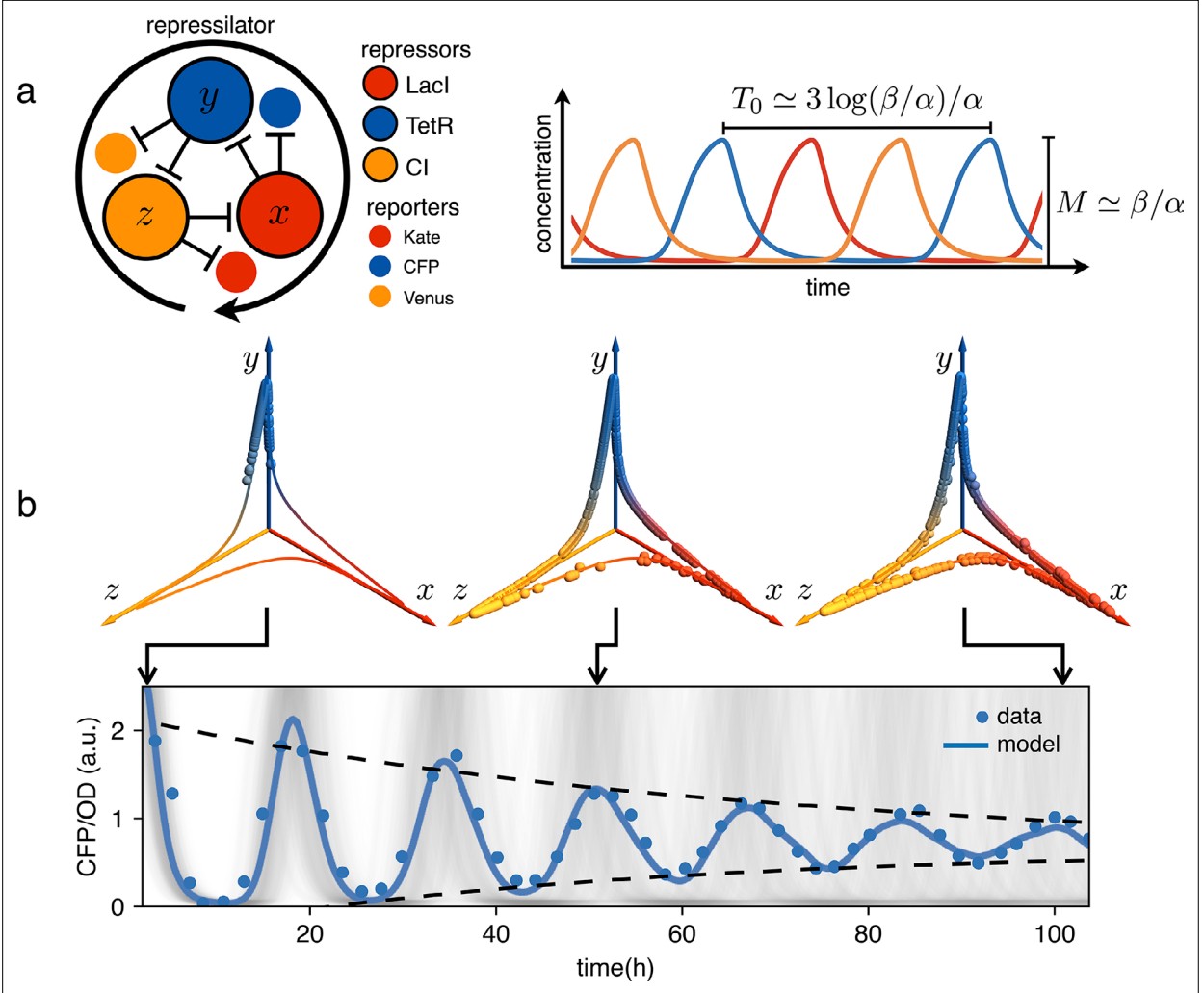

**Figure 1.** Dephasing in a population of repressilators. (**a**) (left) Schematic representation of the repressilator 2.0 plasmid. (right) Simulated time evolution of the three repressors concentrations in the limit cycle of the protein-only model in *Equation 1*. The reported expressions for amplitude and period are derived in the limits $n \to \infty$ and $\beta/\alpha \gg 1$ (see Appendix 1-section 1). (**b**) Concentration of the reporter CFP (TetR) as a function of time for an initially synchronized population of oscillators growing in a multi-well plate. Blue circles represent average over two technical replicates, corresponding errorbars are comparable to symbol size. The shadowed gray curves show simulations of independent oscillators (see Materials and methods) and standard deviation 0.034 h$^{-1}$. Dephasing due to the variability of oscillators periods causes a damping in the mean population signal reported as a blue line. Snapshots of the population ensemble in the 3D concentration space $x$-$y$-$z$ are reported for three instants of time during the simulation, clearly showing progressive dephasing of individual oscillators over the limit cycle trajectory.

on a loop. The original design, using the repressors LacI, TetR and cI in *Escherichia coli*, displayed marked oscillations in single cells, although with large fluctuations in both amplitude and period. This noise has been recently significantly reduced by genetic optimization, resulting in much more robust albeit slower oscillations (*Potvin-Trottier et al., 2016*). This refactored repressilator is usually referred to as repressilator 2.0 (*Riglar et al., 2019*). In this system, the decrease in protein concentration is driven only by growth-related dilution, not by active protein degradation. As discussed in *Elowitz and Leibler, 2000*, a symmetric, protein only model already predicts limit cycle oscillations provided repression is steep enough:

$$\dot{x} = \frac{\beta}{1+z^n} - \alpha x \quad \dot{y} = \frac{\beta}{1+x^n} - \alpha y \quad \dot{z} = \frac{\beta}{1+y^n} - \alpha z \tag{1}$$

where $x, y, z$ are the three repressors concentrations in units of the dissociation constant $K$, the concentration of proteins required to repress a promoter to half maximum, $\alpha$ is a common decay rate set by dilution and $\beta$ is the normalized maximum production rate. In the digital approximation ($n \to \infty$),

when $\beta/\alpha \gg 1$ the resulting dynamics look like relaxation oscillations with amplitude $\beta/\alpha$ and a full cycle period given by $T_0 = 3\log(\beta/\alpha)/\alpha$ (see *Figure 1a* and Appendix 1-section 1). As a result, although individual cells oscillate indefinitely, the natural dispersion of growth rates $\alpha$ results in a dispersion of periods leading to progressive dephasing of individual oscillators as shown by the simulations in *Figure 1b*. In the same figure we report an experimental observation of this damping that we obtained using *E. coli* bacteria carrying the repressilator 2.0 (*Potvin-Trottier et al., 2016*). This system contains three fluorescent reporters whose expression is regulated by the same three transcription factors of the repressilator, so that the reporters' concentration follows that of the repressor transcribed by the same promoter (see *Figure 1a*). The cultures were initially synchronized chemically with IPTG, which deactivates the LacI repressor eliminating the edge between $x$ and $y$ from the network in *Figure 1a*. This new topology admits a stable fixed point where the concentrations of $x$ and $y$ are maximal while $z$ goes to zero. Next, the cultures were maintained in exponential growth phase by periodic dilutions in a multiwell plate while the population-averaged fluorescence signal was monitored using a plate reader (see Materials and methods). For clarity reasons, we will often report only the fluorescence signal from the CFP reporter that indirectly quantifies the concentration of the TetR repressor or $y$ in our simple model. *Figure 1b* shows that when starting from the synchronized state, the population signal from the CFP reporter displays high contrast oscillations with a period of 16 hr and an amplitude that is reduced to a half after about 2.5 periods. Phase drifts are very common in natural biological oscillators. Small couplings within a population of oscillators can give rise to a globally synchronized state (*Pikovsky, 2001*), as in the case of cardiac pacemaker cells (*Mirollo and Strogatz, 1990*) or quorum sensing coupled genetic oscillators (*Danino et al., 2010*; *Prindle et al., 2011*). Other genetic oscillators, like circadian clocks, rely instead on a periodic external cue, known as zeitgeber (time-giver), to constantly adjust their phase to that of a common environmental cycle. Sunlight is the predominant zeitgeber of natural circadian clocks (*Golombek and Rosenstein, 2010*). In the context of synthetic biology, light is a particularly versatile input for controlling the state of genetic circuits and programming gene expression in space and time with much greater precision than chemical signals (*Levskaya et al., 2005*; *Tabor et al., 2009*; *Olson et al., 2014*; *Baumschlager and Khammash, 2021*). Optical signals can also be multiplexed through spectral (*Schmidl et al., 2014*; *Fernandez-Rodriguez et al., 2017* or amplitude modulation *Benzinger et al., 2022*). A growing number of optogenetics tools have been recently applied in prokaryotes to control different aspects of bacterial physiology (*Liu et al., 2018*, such as growth *Davidson et al., 2013*; *Milias-Argeitis et al., 2016*; *Gutiérrez Mena et al., 2022*, antibiotic resistance *Sheets et al., 2023*, motility *Frangipane et al., 2018*; *Zhang et al., 2020* and adhesion *Jin and Riedel-Kruse, 2018*). A widely used optogenetic system is the light-switchable two-component system, CcaS-CcaR from *Synechocystis* PCC 6803 (*Tabor et al., 2011*). CcaS senses green light and phosphorilates the response regulator CcaR. Phosphorylated CcaR binds to the promoter $P_{cpcG2-172}$, activating gene expression. Red light reverts CcaS to the inactive state and shuts down transcription from $P_{cpcG2-172}$. This system was optimized for controllable gene expression with a high dynamic range in *E. coli* (*Schmidl et al., 2014*; *Ong and Tabor, 2018*).

Here, we show that integrating an optogenetic module in the repressilator circuit enables the use of light to synchronize, entrain, and detune oscillations in gene expression within single cells or entire populations. We employ the CcaS-CcaR light-inducible system to express one of the repressilator proteins, resulting in a novel four-node optogenetic network named the 'optorepressilator'. With this modification, light induces precise phase adjustments among synthetic genetic clocks within individual cells, leading to persistent population-wide oscillations. These oscillations mantain a constant phase relation to the external light cue that can act as a zeitgeber.

We show that a population of these synthetic oscillators can be synchronized through transient green light exposure or be entrained via a sequence of short pulses, sustaining indefinite oscillations. Additionally, we explore the system's response to detuned external stimuli, revealing multiple synchronization regimes.

## Results

### Optorepressilator: a light-controllable repressilator

In the optorepressilator (*Figure 2a*), LacI proteins are produced by two genes. We indicate with $x$ the normalized concentration of the repressilator's LacI, transcribed via the $\lambda$ promoter and repressed

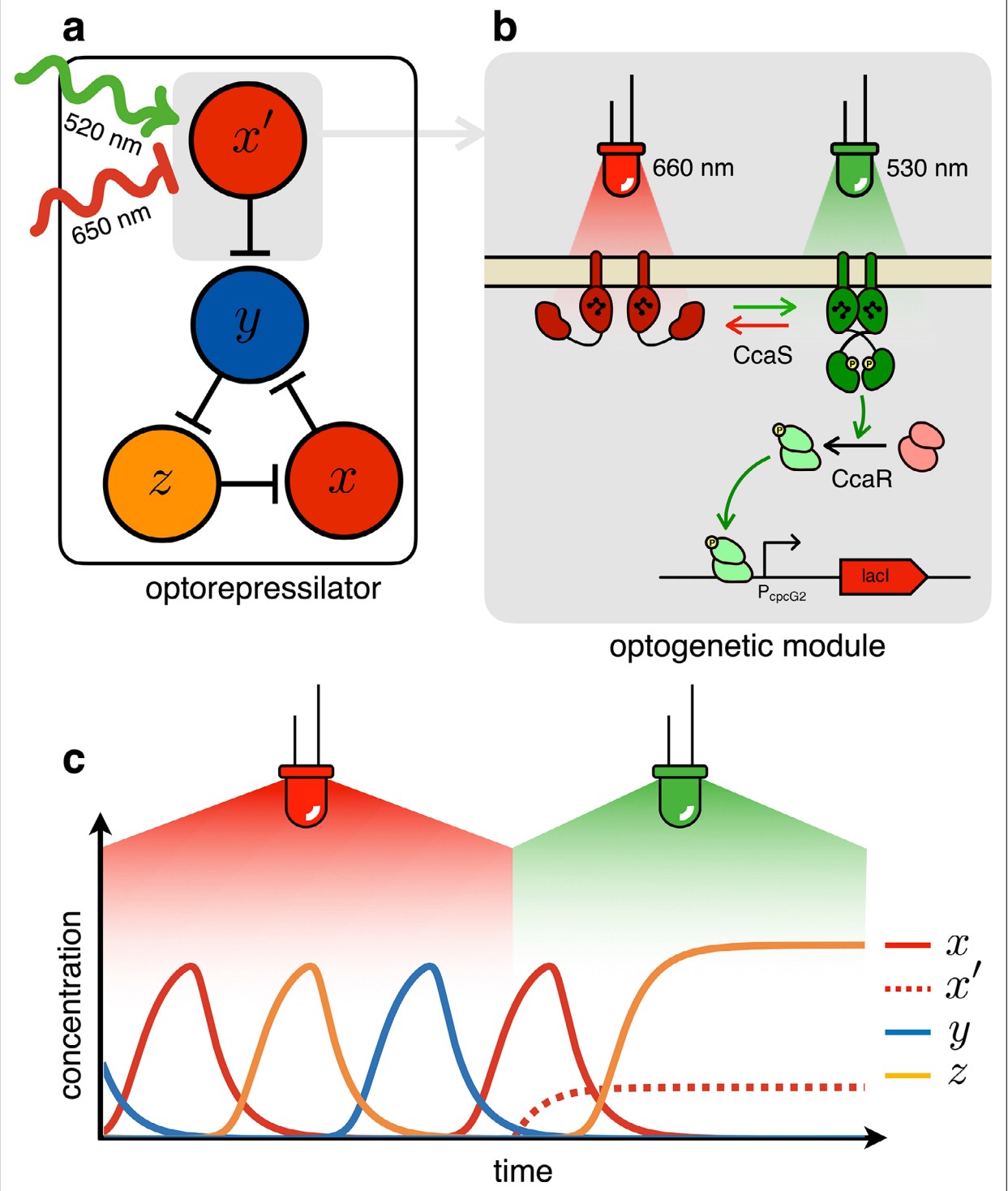

**Figure 2.** The optorepressilator: a light-controllable genetic oscillator. (**a**) Schematic illustration of the optorepressilator circuit. Green and red light respectively promote and repress the production of one of the three transcription factors in the repressilator (LacI). (**b**) Working mechanism of the optogenetic module. A light-driven two-component system controls the production of an additional copy of *lacI*. The optogenetic system consists of the membrane-associated histidine kinase CcaS and its response regulator CcaR. Absorption of green light increases the rate of CcaS autophosphorylation and phosphate transfer to the transcription factor CcaR. Phosphorylated CcaR promotes transcription of an additional copy of the LacI gene from the promoter $P_{cpcG2-172}$. Red light reverts CcaS to the inactive state and shuts down transcription from $P_{cpcG2-172}$. (**c**) Ideal optorepressilator's dynamics according to *Equation 2*. The system oscillates unperturbed under red light, but collapses to a fixed point under green light when the extra $x'$ represses $y$.

by $z$ (cI), and with $x'$ the concentration of LacI proteins transcribed via the light-inducible promoter $P_{cpcG2-172}$ (*Schmidl et al., 2014*, *Figure 2b*). $x$ and $x'$ add up to repress protein $y$, while a light-dependent production rate drives $x'$ dynamics:

$$\dot{x} = \frac{\beta}{1 + z^n} - \alpha x, \quad \dot{y} = \frac{\beta}{1 + (x + x')^n} - \alpha y, \quad \dot{z} = \frac{\beta}{1 + y^n} - \alpha z, \quad \dot{x}' = \beta' - \alpha x' \quad (2)$$

where $\beta'$ is a light dependent production rate. It is important to note that the dynamics of $x'$ is decoupled from all other repressors and only determined by light. In particular, under steady illumination conditions $x' = \beta'/\alpha$. If the expression of the light-inducible LacI is adjusted to have a dynamic range containing the dissociation constant $K$ then we might be able to optically switch the system from limit cycle oscillations under red light to a fixed point under green light (*Figure 2c*). In the digital approximation, the limit cycle is broken when $\beta'/\alpha > 1$ and the system collapses to a fixed point with coordinates $x_0 = 0, y_0 = 0, z_0 = \beta/\alpha$.

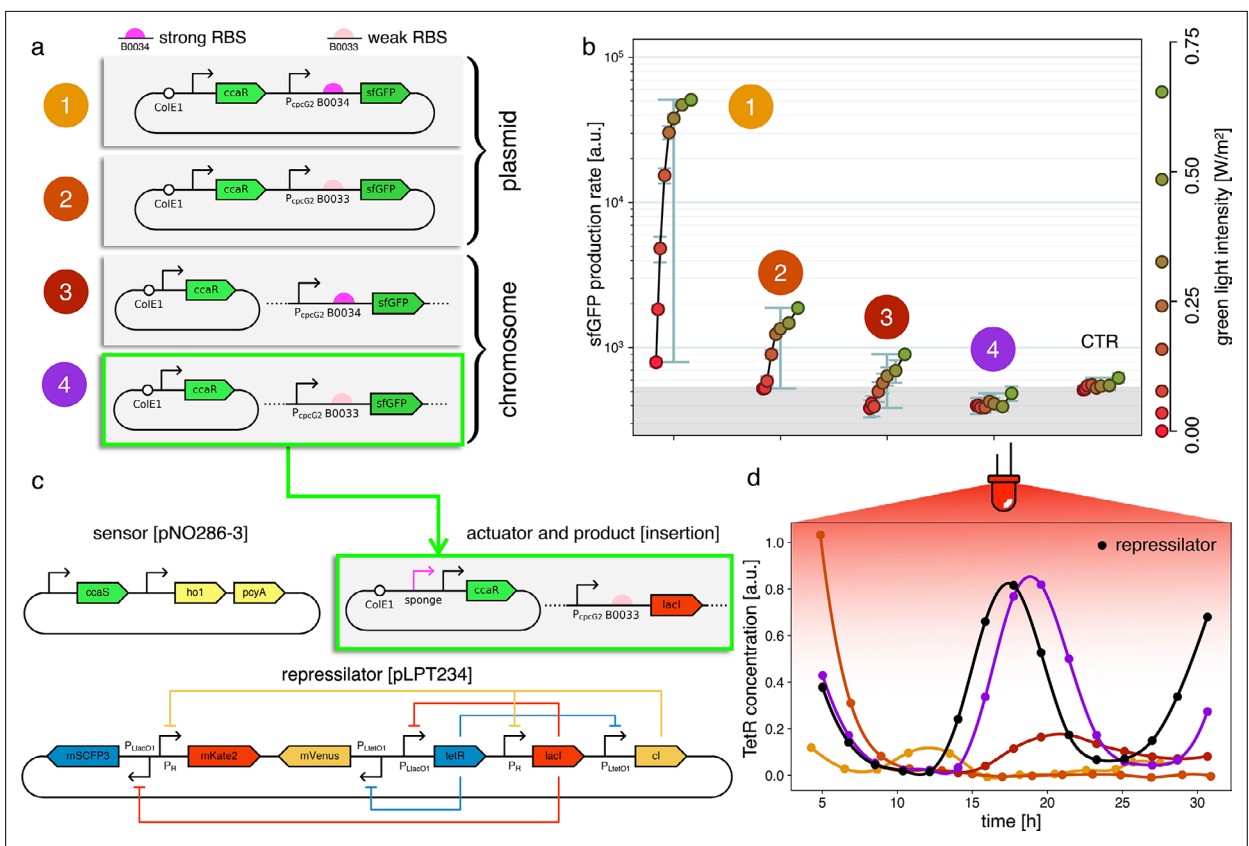

**Figure 3.** Fine-tuning gene expression in the optorepressilator circuit. (**a**) To tune expression from the optogenetic module we varied the strength of RBSs and gene copy number by placing the output gene (sfGFP) either on a plasmid or in the chromosome. (**b**) Light-controlled expression levels of the sfGFP reporter from the four constructs in (**a**) and a negative control consisting of DHL708 with plasmids pNO286-3 and pSR58-0 (optogenetic plasmids without sfGFP cassette). For each curve, the samples were exposed to a fixed level of red light (intensity 0.74 W/m²) while increasing green light intensity, as indicated by the dots' colors (color reference on the right). The main error bars represent the dynamic range of each sample. The smaller error bars represent the standard deviation of individual light conditions between replicates made in three separate days. The gray shaded area represents the autofluorescence background. Light intensities and raw fluorescence data are in *Appendix 1—figure 6*. (**c**) Scheme of the final optorepressilator circuit (see Materials and methods for details). (**d**) Time evolution of TetR reporter (CFP) concentration in a population of IPTG synchronized cells growing under red light. Black points from the original repressilator 2.0 display marked oscillations. Colored lines correspond to the four constructs in a with sfGFP replaced by LacI and the addition of the sponge promoters as in pSpongeROG. Circles represent data, where each dot is the average of two replicates with error bars comparable to marker size, while lines are spline interpolations. Only the purple line, corresponding to the system in (**c**), shows oscillations comparable to those of the original repressilator. The difference in period can be explained by the difference in growth rate of the two strains, as shown in *Appendix 1—figure 4*.

## Fine-tuning optogenetic expression

Estimates for the LacI dissociation constant $K$ vary between 0.01–0.1 nM (*Ozbudak et al., 2004*). Even taking into account the fact that the repressor plasmid is present in more than one copy (~ five copies), this suggests that a small number of LacI proteins may already disrupt the limit cycle. Therefore, controlling the leakage of the promoter under the red light is crucial. The optogenetic module consists of two components, named as sensor and actuator in *Figure 3c*. In order to fine-tune the expression of LacI, we created four different ranges of gene expression in which both transcription and translation were modulated on two levels. Translation was controlled by substituting the ribosome binding site (RBS) (BBa_B0034) in the original plasmid with a weaker RBS (BBa_B0033) from the same IGEM Community collection. Transcription was modulated through gene copy number by moving the light-driven gene expression cassettes from the plasmids to the genome. Combined with the sensor and actuator plasmids, these constructs resulted in four versions of the optogenetic module (*Figure 3a*) with different expression ranges. We characterized these expression ranges as a function of incident light (*Figure 3b*) using sfGFP as fluorescent reporter and a custom-made light-addressable multiwell plate (see *Appendix 1—figure 3*). By comparing expression at maximum green level, we can estimate that RBS substitution leads to a 27-fold decrease in reporter production (*Figure 3b*), while relocation of the light-driven cassette to the genome reduces expression by 57-fold. The combination of weak RBS and genome insertion results in a gene expression range below the autofluorescence background (estimated combined fold reduction of 1500). We then replaced the sfGFP gene with LacI in all four versions of the optogenetic module and transformed the repressilator plasmid (pLPT234) in each of the four strains. As a first step, we verified that the oscillations of the limit cycle were preserved under red light. To this end, we first chemically synchronized the oscillations with IPTG and then monitored the concentration of TetR ($y$) reported by CFP. Cultures were maintained in exponential phase by periodic dilutions in multiwells under constant red light. We found that only the strain with the lowest expression level (represented in *Figure 3c*) oscillates with the same amplitude as the control strain containing only the repressilator plasmids (*Figure 3d*). The slight difference in period can be explained by the presence of additional plasmids in the optorepressilator strain, which results in a lower growth rate. As found in the digital approximation, the repressilator period is mainly controlled by the inverse growth rate (see *Figure 1a* and *Appendix 1—figure 9*) meaning a lower growth rate results in a longer oscillation period. When we normalize the time with the growth rate the two oscillations overlap nicely (*Appendix 1—figure 4*). In contrast, no clear oscillations were observed in the other three strains, where LacI leaking from the red-repressed promoter destroys the limit cycle and collapses all cells in a fixed point where the repressilator protein cI is high and TetR is low (see Appendix 1-section 5).

## Optogenetic synchronization

Having verified that the system oscillates under red light, we then checked whether green light can synchronize a population of optorepressilators. *Figure 4a* shows the population signal of CFP fluorescence reporting the concentration of TetR ($y$) in multiwell cultures. The cultures were constantly kept under red light for the first 40 hr. Although individual cells oscillate, their phases are randomly distributed so that the average population signal is constant. At $t = 40$ h we switch from red to green for 12 hr, and CFP fluorescence decays to zero. This is expected when extra LacI is produced by the optogenetic module, repressing both TetR and CFP. When the population signal decays completely, all the cells are stuck in the same fixed point ($x_0 = 0, y_0 = 0, z_0 = \beta/\alpha$ in the model), so that when switching back to red, they start oscillating in synchrony. This was also confirmed by single-cell data from a mother-machine microfluidic chip (*Figure 4b*). Under the same light protocol as in *Figure 4a*, most of the channels blink in unison after exposure and removal of synchronizing green light (*Video 1*).

## Optogenetic entrainment

Both plate reader and mother-machine experiments showed that cells carrying the optorepressilator system could be synchronized by transient light exposure. However, oscillations in the population signal were progressively dampened again by the dispersion of growth rates and thus of individual clocks' periods. This was particularly evident in the mother-machine experiment, where the growth rate variability is larger. Natural genetic clocks are able to counteract period dispersion by exploiting an external periodic stimulus to advance or delay their phases. For example, it was shown that a

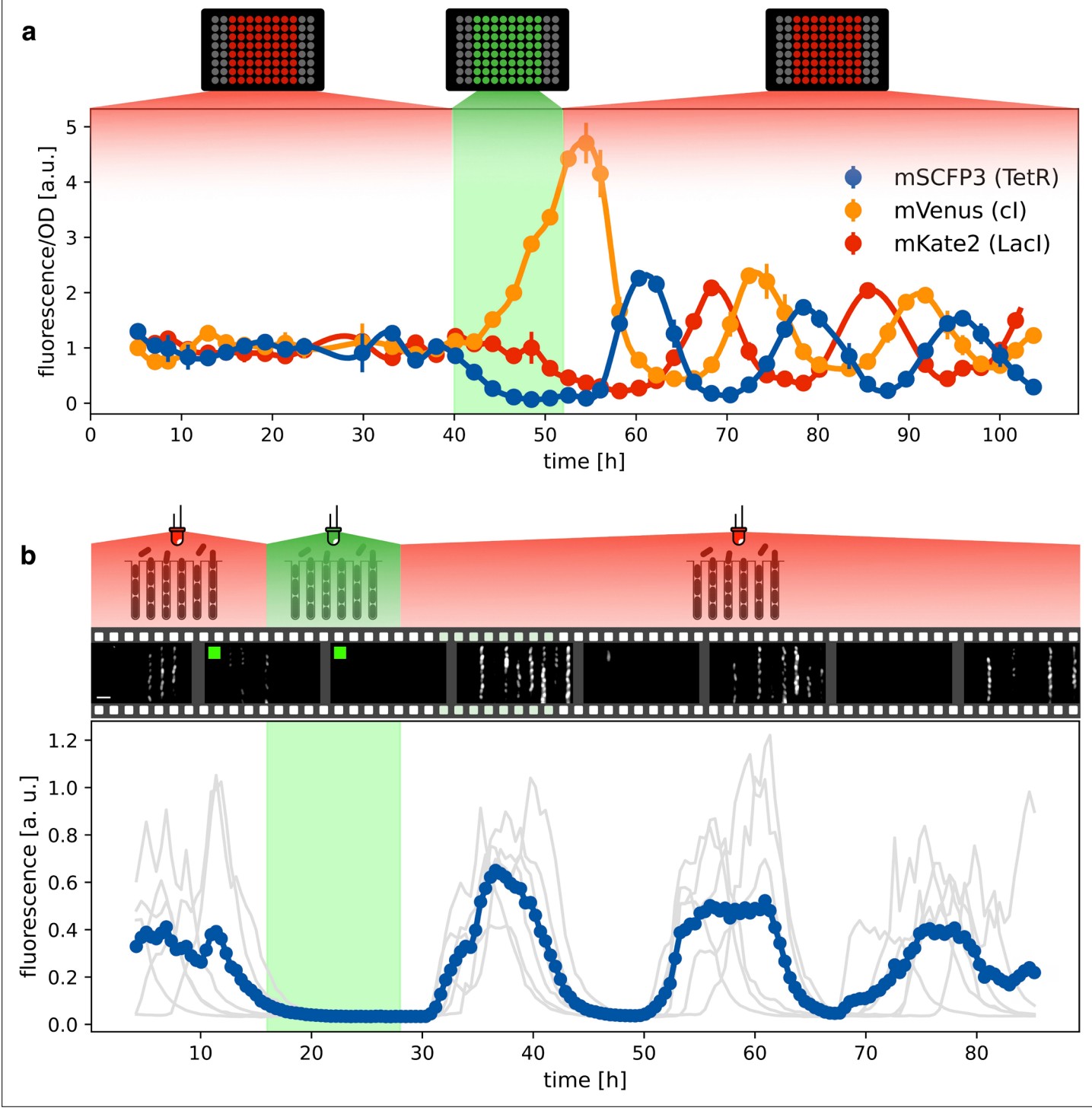

**Figure 4.** Optical synchronization. (**a**) Time evolution of mSCFP3(CFP), mVenus and mKate2 signals from a population of exponentially growing cells in multiwell plate. Red and green shaded areas represent the illumination protocol. Dots reports data for optorepressilator cells and clearly shows the appearance of synchronous oscillations after transient illumination with green light, each marker is the average of two replicates and the error bar is data range. (**b**) Single cell data from a mother machine experiment employing a similar light protocol as in (**a**) (*Video 1*). The gray curves represent the concentration of CFP for individual cells growing in different channels, while the blue curve is their average. Snapshots above the plot report fluorescence imaging of bacteria in the microfluidic chip, centered at the corresponding time point on the time axis below. Scale bar is 5 μm.

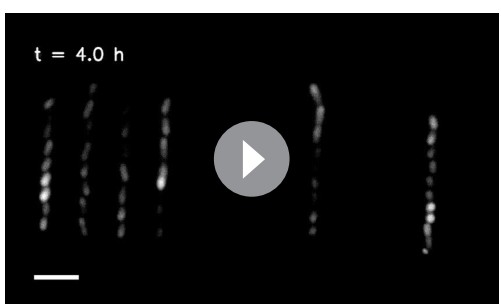

t = 4.0 h

**Video 1.** Optical synchronization in mother machine. The snapshots of CFP fluorescence were acquired every 9 minutes and the video is played at 32 frames/s. The presence of a green square over the image indicates that at that time point the bacteria are exposed to green light stimuli provided by the Thorlabs LED M530L4. The fluorescence of each mother cell and the mean fluorescence of all mother cells are plotted in *Figure 4b*. Scale bar is 5 µm.

https://elifesciences.org/articles/97754/figures#video1

1-hr exposure to light at bedtime can delay the human circadian rhythm by about two hours, while exposure to the same intensity of white light after waking up can advance our schedule by a little less than half an hour (*St Hilaire et al., 2012*). Interestingly, the optorepressilator model in *Equation (2)*, despite its simplicity, displays a very similar behavior. In *Figure 5a*, we report the phase shift $\Delta\phi$ produced by a single pulse (with a duration of 2 hr and intensity $\beta' = 80 \text{ h}^{-1}$ fixed for all the simulations) as a function of the pulse arrival phase $\phi$. This type of curve is often referred to as a phase response curve (PRC) and is obtained here by numerical integration of (2) for a single optorepressilator. The first thing we notice is that positive pulse arrival phase $\phi > 0$ results in a negative phase shift $\Delta\phi < 0$ that delays subsequent oscillations, while a negative $\phi$ results in a phase advance ($\Delta\phi > 0$). This behavior can be understood at least qualitatively by simple considerations. When a pulse arrives before $y$ reaches the maximum ($\phi < 0$), a burst of $x'$ is produced

(*Figure 5b*) that may trigger an earlier decay of $y$ if $\beta'/\alpha > 1$ and the pulse duration $\tau$ is long enough. This results in an advanced peak, or equivalently a positive phase shift, which is greater the earlier the light pulse arrives. Conversely, when the pulse arrives while $y$ is decaying the $x'$ burst will prolong the $y$ decay until the moment when $x'$ falls below 1. This results in a delayed peak, or a negative phase shift that increases in magnitude as $\phi$ increases (*Figure 5b*). The exact shape of the PRC will depend on light intensity through $\beta'$ and pulse duration $\tau$ as shown in *Appendix 1—figure 7*.

The phase response curve turns out to be a very useful concept to predict the system's response to a periodic external input. For example, if the system is exposed to a periodic train of light pulses, the phase response curve can be used to predict the sequence of phases $\phi_n$ at which subsequent pulses will arrive (*Granada et al., 2009*).

$$\phi_{n+1} = \phi_n + \frac{T_L}{T_0} - 1 + \Delta\phi(\phi_n) \tag{3}$$

where phases go from 0 at the expression peak of the $y$ protein, to 1 when a full cycle is completed at the next peak. The fixed points $\phi^*$ of this mapping are those for which $\phi_{n+1} = \phi_n$ and hence:

$$\Delta\phi(\phi^*) = 1 - \frac{T_L}{T_0} \tag{4}$$

The fixed points where the phase response curve has a negative slope satisfying the condition $-2 < d\Delta\phi/d\phi < 0$, represent stable entrained states (*Granada et al., 2009*) for which the system oscillates at the same frequency of the external light signal. So when the system is exposed to a train of external pulses of period $T_L$ we should look for the phase $\phi^*$ for which *Equation 4* is satisfied, and if the phase response curve has a negative slope larger than –2, we can predict that the system will be entrained with a stable phase difference $\phi^*$ between light pulses and the peaks of CFP ($y$) protein.

Let us now consider an ensemble of optorepressilators with distributed natural periods $T_0$ and subjected to a train of light pulses with period $T_L$. Based on the previous discussion, each optorepressilator will be entrained by the external periodic signal but with a specific phase difference satisfying *Equation 4*. The population signal will then show entrained oscillations representing the average over individual oscillators having the same period but maintaining different phase differences with the external signal.

From this simplified theoretical discussion, we deduce that if real optorepressilators had the predicted phase response curve, then it should be possible to use a train of light pulses as a zeitgeber capable of producing long-term oscillations in gene expression at the population level. We first

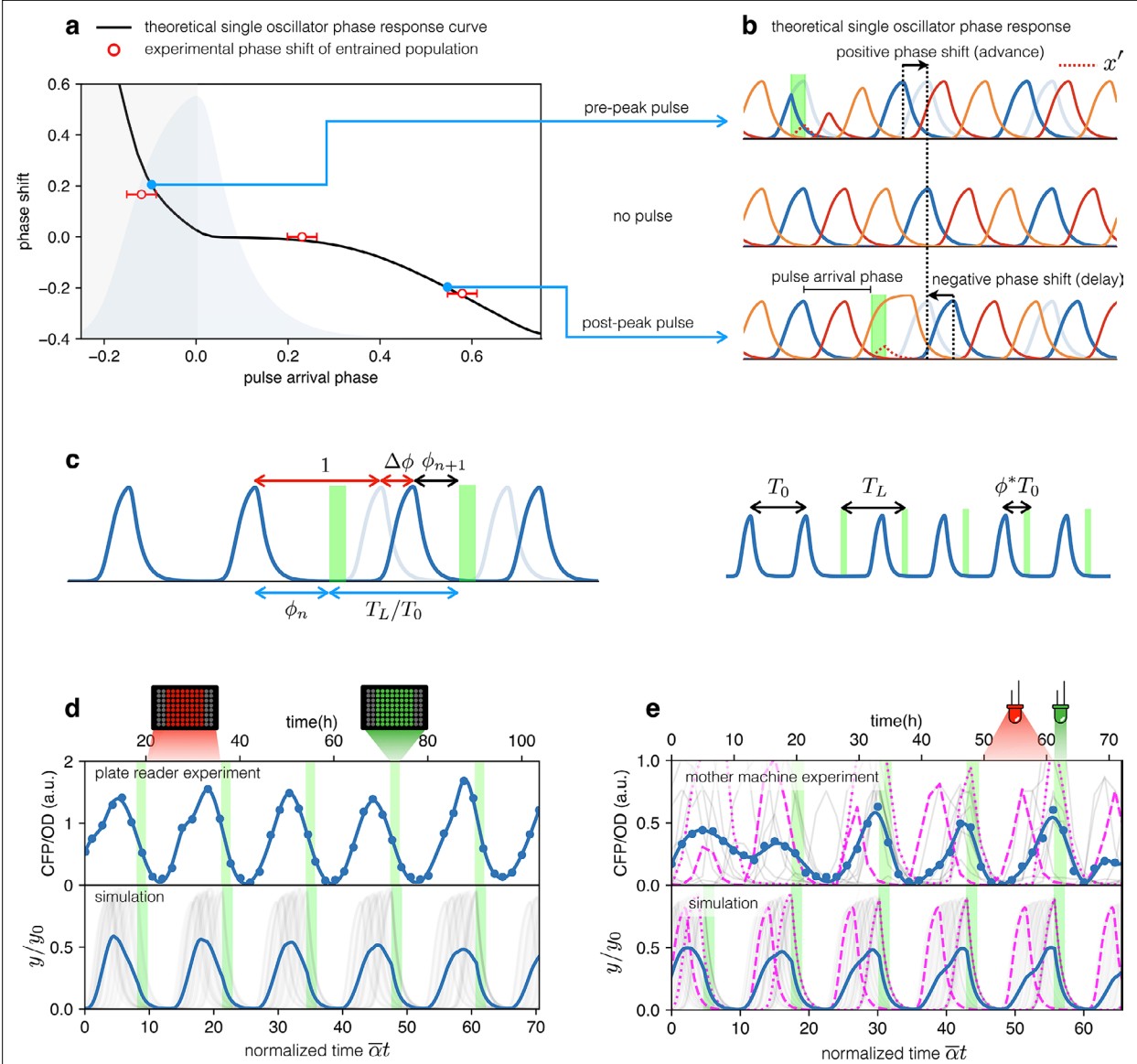

**Figure 5.** Optical entrainment. (**a**) Phase shift as a function of pulse arrival phase. (**b**) Numerical simulations of **Equation (2)** illustrating the effect of pulse arrival phase on the phase of optorepressilator oscillations. (**c**) Graphical illustration of the parameters used to describe the phase shift. (**d**) (top) CFP signal from a population of cells growing exponentially in 96-well plates that are exposed to a train of periodic pulses of green light. The population signal displays undamped oscillations demonstrating optical entrainment. (bottom) Numerical simulation of an ensemble of optorepressilators evolving according to **Equation (2)** and with distributed growth rates (same parameters as in **Figure 1b**). Pulses have a duration of $\tau = 2$ h and a relative period of $T_L/T_0 = 1$. Gray lines are individual oscillators while blue is the population average. (**e**) (top) Gray lines are CFP signals from single cells growing in a mother machine under periodic pulsed illumination (**Video 2**). The blue curve is the average. (bottom) Numerical simulation with the same parameters as in (**d**) but a larger dispersion of growth rates and a slightly shorter relative period $T_L/T_0 = 0.97$ was required to match experiments above. In both data and simulation the magenta dotted curve highlights a slow oscillator (low growth rate and larger amplitude) entrained with a phase such as to receive the light pulse on the rising edge to anticipate the next oscillation. Magenta dashed lines highlight faster oscillators whose phase is such to receive the pulse on the decaying edge to delay the following oscillation.

demonstrate this by monitoring the population signal from CFP (reporting TetR or $y$ in the model) in multiwell cultures under constant red illumination (6.82 W/m²) interrupted by green light pulses (5.64 W/m²) of duration 2 h with period $T = 18$ h. **Figure 5d** shows high contrast and undamped oscillations to be contrasted to the damped ones in **Figure 1b** and **Figure 4a**. The bottom panel in **Figure 5d** shows the result of a numerical simulation with the same parameters as in **Figure 1b** and the addition of a periodic light stimulation, with period $T_L/T_0 = 1$. Simulation panel also displays in

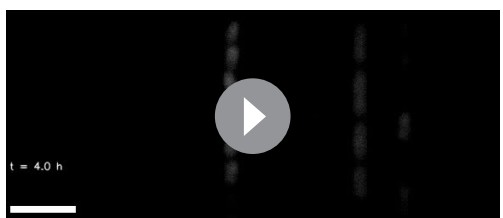

**Video 2.** Optical entrainment in mother machine. The snapshots of CFP fluorescence were acquired every 9 minutes and the video is played at 32 frames/s. The presence of a green square over the image indicates that at that time point the bacteria are exposed to green light stimuli provided by the Thorlabs LED M530L4. The fluorescence of each mother cell and the mean fluorescence of all mother cells are plotted in *Figure 5e*. Scale bar is 5 μm.

https://elifesciences.org/articles/97754/figures#video2

gray the signals from individual optorepressilators before performing the average (in blue). These results are also confirmed by mother machine observations reported in *Figure 5e* (*Video 2*). For the simulations in the lower panel of *Figure 5e*, all parameters remained the same as in *Figure 5d* with the exception of the period of the light pulses ($T_L/T_0 = 0.97$) and the standard deviation of the growth rate distribution, which was increased from 0.034 h$^{-1}$ to 0.071 h$^{-1}$ to better reproduce the experimental observations in the mother machine. A narrower distribution of growth rates in multiwell cultures could be due to the fact that faster growing bacteria outperform slower-growing cells in a competing environment, unlike in the mother machine where there is no competition between cells in different channels. The theoretical discussion above predicts that the individual oscillators will each find their own stable phase with respect to external pulses. The slower ones will arrange themselves to receive the pulse on the rising edge of the peak to reduce their period, while the faster ones will have the pulse arriving after the peak to increase their period. We confirm this in both experiments and simulations

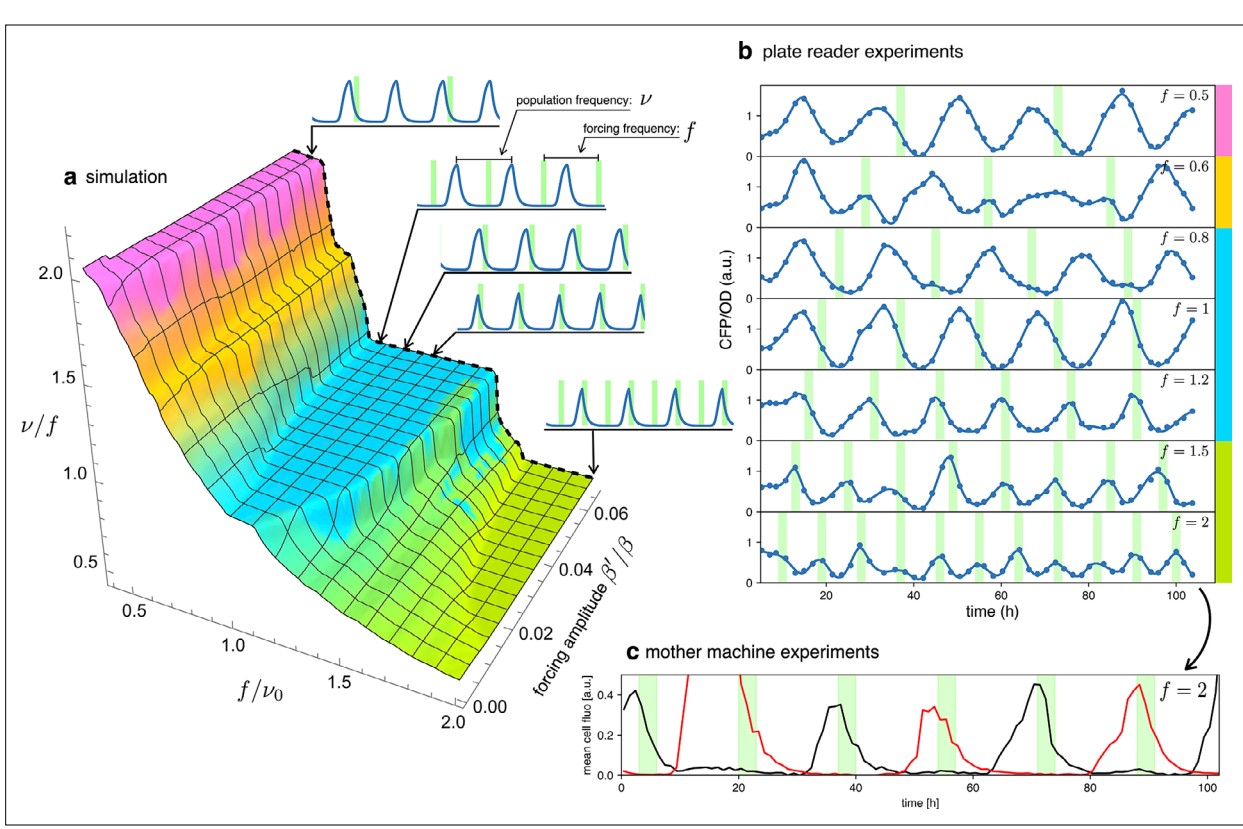

**Figure 6.** Detuning. (**a**) 3D surface representing the ratio between the actual frequency of a single forced optorepressilator and the frequency of the forcing signal, as a function of the forcing frequency and amplitude. The colors of the plateaus (Arnold's tongues) highlight different regions of global synchronization in the frequency/amplitude plane. (**b**) Time evolution of the average CFP concentration (blue curve) in a plate reader experiment, for different frequencies of green light pulses (green bars). The color bars at the right of the plot indicates the corresponding region on the surface in (**a**). (**c**) Mother machine experiment revealing two distinct cells oscillating at half the signal frequency but peaking on alternate pulses of incoming light. This explains why for $f = 1.5$ and $f = 2$ in (**b**) we observe a population mean oscillating with the same frequency of the forcing instead of the $\nu/f = 0.5$ value predicted for the green tongue in (**a**).

in *Figure 5e*, where the individual traces of a slow (higher amplitude) and a fast (lower amplitude) oscillator have been highlighted with dotted and dashed lines, respectively.

## Detuning

In an entrained population, most oscillators are detuned from their natural frequency to that of the external signal. In this section, we will explore detuning by starting with numerical simulations of our simple model in *Equation 2* and then comparing them with long-term observations of detuned optorepressilators in a plate reader experiment. *Figure 6a* shows a 3D surface representing the ratio of the actual optorepressilator frequency $\nu = 1/T_0$ over external frequency $f = 1/T_L$ as a function of frequency and amplitude of the external signal $\beta'$. The $\nu$ value is obtained as the frequency of the main peak in the power spectrum of the simulated data. The plateaus in this graph are related to the so-called Arnold's tongues (*Pikovsky, 2001*), that is regions of synchronization in the frequency/amplitude plane. On the cyan plateau, the system oscillates with the same frequency as the external signal, while there are also higher order plateaus with a fractional value of the ratio $f/\nu_0$ (see *Appendix 1— figure 8*). On the cyan tongue, when the system is detuned to higher frequencies, its phase will be such that it receives the light pulse on the rising edge of the $y$ protein. When detuned to lower frequencies, however, it shifts to receive pulses on the falling edge of $y$. When detuning to lower frequencies, the system goes to an higher order synchronization tongue where it oscillates at twice the frequency of the external signal. Similarly, forcing to higher frequencies moves the system to fractional order tongues, such as the green plateau in the plot, in which the system oscillates at half the external frequency. All these regimes are found in experiments in which we maintain cultures of optorepressilators in exponential phase within multiwell plates that are independently addressed with light signals of different frequencies (*Figure 6b*). It is also remarkable to observe how the relative phase between the system and the external signal evolves within the cyan tongue, as predicted by the model. More quantitatively, we have extracted the phase difference $\phi^*$ between light pulses and CFP oscillations from the data shown in the three central panels of *Figure 6b* ($f = 0.8, 1, 1.2$). We found that, in accordance with the discussion in the previous section (*Equation 4*), the red points with coordinates ($\phi^*, 1 - T_L/T_0$) fall on the theoretical phase response curve when we choose $\beta' = 80\ \mathrm{h}^{-1}$ (*Figure 5a*).

A discrepancy is apparently found in the green tongue, where we experimentally observe oscillations at the same frequency as the external signal. This is not unexpected, however, when one realizes that although the individual optorepressilators may oscillate at half the external frequency $f$, they can peak at either the even or odd peaks of the external signal. When we average these alternating peaks over the population, we get a small amplitude oscillation at the frequency of the external signal. This is also confirmed by a mother machine experiment where we tried to detune to about twice the natural frequency and observed different channels oscillating at half the signal frequency and with alternating peaks (*Figure 6c*).

## Discussion

Inspired by the light entrainment of natural circadian rhythms, we designed the optorepressilator, a four-node synthetic oscillatory network that can lock its phase to an external periodic light signal. By integrating an optogenetic module into a repressilator circuit, we demonstrate, theoretically and with experiments, that light can synchronize, entrain, and detune oscillations within single cells or an entire population. We find that this four-node network shows a phase-sensitive response to light pulses. Depending on the arrival time, a pulse can either delay or advance the clock as in many natural circadian clocks. As a result, our system can be entrained by periodic light pulses, even when the forcing period deviates from the natural frequency. Using a combination of optogenetic experiments ranging from the macroscopic population scale to the microscopic scale of single cells, we show that the entrainment mechanism is robust and can be understood quantitatively by a simple protein-only model. The entrainment of biological clocks can provide new insights into oscillation mechanisms and, at the same time, practical means to control biological rhythms. Through periodic modulation of chemical signals within microfluidic environments, entrainment of natural biological clocks ranging from the embryonic segmentation clock (*Sanchez et al., 2022*) to the NF-kB system regulating immune response (*Heltberg et al., 2016*) has been previously achieved. However, the complexity of the underlying genetic networks often precludes a deeper mathematical understanding of the

fundamental nature of the entrainment phenomenon. Synthetic oscillators provide an ideal platform for studying entrainment in engineered minimal systems where a close comparison with relatively simple models is possible (*Mondragón-Palomino et al., 2011*). Our minimal optorepressilator model can provide many theoretical predictions, often in analytical form within the digital approximation ($n \to \infty$). It explains correlations between amplitude and period of oscillations (*Appendix 1—figure 9*), or how robust oscillations are against leakage (Appendix 1-section 5). Moreover an approximate form for the phase response curve is derived (Appendix 1-section 7), providing a quantitative physical insight into the mechanisms that produce delayed or advanced oscillations and on how pulse characteristics affect the system's response.

From a more applied perspective, the optorepressilator enriches the toolbox of genetic modules by adding a light-synchronizable clock that could be used to program precise oscillatory patterns of gene expression. When compared to chemical signaling, the use of light to control and coordinate the functioning of synthetic networks offers clear advantages in terms of spatial and temporal modulation. Light cues can be applied and removed on the spot, without waiting for media substitution (*Zhao et al., 2018*). In addition, using spatial light modulators it could be possible to address spatially separated subpopulations within a common environment to oscillate with distinct periods and phases. Optical synchronization of genetic oscillators can be easily implemented in a wide range of experimental settings like liquid cultures, agar plates or microfluidics. Light, for example, can penetrate bioreactors and orchestrate gene expression in a population of uncoupled oscillators. Moving to infrared inducible optogenetic systems (*Chernov et al., 2017*), optical signals could also penetrate biological tissues for in vivo applications (*Riglar et al., 2019*).

## Materials and methods

**Key resources table**

| Reagent type (species) or resource | Designation | Source or reference | Identifiers | Additional information |
|---|---|---|---|---|
| Strain, strain background (*Escherichia coli*) | DHL708 | *Potvin-Trottier et al., 2016* | Addgene_98417 | |
| Recombinant DNA reagent | pLPT234 (plasmid) | *Riglar et al., 2019* | RRID:Addgene_127855 | |
| Recombinant DNA reagent | pLPT145 (plasmid) | *Potvin-Trottier et al., 2016* | RRID:Addgene_85527 | |
| Recombinant DNA reagent | pNO286-3 (plasmid) | *Ong and Tabor, 2018* | RRID:Addgene_107746 | |
| Recombinant DNA reagent | pSR58.6 (plasmid) | *Schmidl et al., 2014* | RRID:Addgene_63176 | |
| Recombinant DNA reagent | pKD46 (plasmid) | *Datsenko and Wanner, 2000* | | |
| Recombinant DNA reagent | pKD4 (plasmid) | *Datsenko and Wanner, 2000* | RRID: Addgene_45605 | |
| Commercial kit | In-Fusion HD Cloning | Takara Bio | | |

### Strains and plasmids

A list of the strains and plasmids used is in *Appendix 1—table 1* and *Appendix 1—table 2*, along with plasmids maps and their description (*Appendix 1—figures 10 and 11*). Genome insertion of light-driven cassettes was performed using lambda-red recombination (*Datsenko and Wanner, 2000*) targeting the *attB* genome site. The starting strain was *E. coli* DHL708 (a gift from Johan Paulsson, Addgene plasmid # 98417; *Potvin-Trottier et al., 2016*) for all insertions. All the recombination cassettes were amplified from modified versions of plasmid pKD46 (*Datsenko and Wanner, 2000*, *Appendix 1—table 2*). In addition to kanamycin antibiotic resistance, the recombination cassettes contained the desired protein gene (LacI, sfGFP) controlled by the light-driven promoter $P_{cpcG2-172}$.

Plasmid cloning was performed using In-Fusion cloning (Takara Bio) for large rearrangements and Q5 Site-Directed Mutagenesis Kit (New England BioLabs) for small modifications that is RBS changes. The final optorepressilator system is composed of: the strain MCC0233 obtained inserting $P_{cpcG2-172}$ (the light-driven promoter), a weak RBS and LacI open-reading frame in the genome of DHL708; pNO286-3 (a gift from Jeffrey Tabor, Addgene plasmid # 107746; http://n2t.net/addgene:107746; RRID:Addgene_107746) *Ong and Tabor, 2018*, plasmid with mini-CcaS light sensor and the enzymes for the phycocyanobilin chromophore; pLPT234 (a gift from Johan Paulsson, Addgene plasmid # 127855; http://n2t.net/addgene:127855; RRID:Addgene_127855) *Riglar et al., 2019*, plasmid with

the core repressilator circuit and reporters; pSpongeROG, a modified version of plasmid pSR58.6 (a gift from Jeffrey Tabor, Addgene plasmid # 63176; http://n2t.net/addgene:63176; RRID:Addgene_63176) *Schmidl et al., 2014* added with the sponge from plasmid pLPT145 (a gift from Johan Paulsson, Addgene plasmid # 85527; http://n2t.net/addgene:85527; RRID:Addgene_85527) *Potvin-Trottier et al., 2016* and without the light-driven gene expression cassette (sfGFP).

## Plate reader experiments to follow circuit oscillations

Bacterial strains were grown from glycerol stock for 16 hr in LB with appropriate antibiotics (see *Appendix 1—table 3* for antibiotic working concentration), in a falcon tube exposed to saturating 660 nm red light in a shaker incubator (200 rpm) at 37 °C. Cultures were diluted $1:10^6$ approximately 8 hr before the beginning of the experiment in imaging medium (*Potvin-Trottier et al., 2016*) with antibiotics under red light, and 1 mM IPTG was added in this interval if samples had to be chemically synchronized. Samples were transferred in duplicates to a 96-well plate (Greiner GR655096), and we measured OD600 and fluorescence for sCFP, mVenus and mKate2. When OD600 reached approximately 0.8 for the first time, the samples were diluted 1:4 in new wells with imaging medium plus antibiotics. Every two hours, we measured OD600 and fluorescence and diluted about 1:4 in fresh culture medium plus antibiotics to bring the cultures back to an OD of 0.2 and keep the bacteria under exponential growth conditions. OD600 and fluorescence were measured with a TECAN Infinite M Nano +plate reader warmed to 37°C. The parameters to detect fluorescence were: mSCFP3 excitation 433 (±9) nm emission 474 (±20) nm; mVenus excitation 500 (±9) nm emission 540 (±20) nm; mKate2 excitation 588 (±9) nm emission 633 (±20) nm. Between measurements, the well plate was kept in a shaking incubator (100 rpm) at 37°C and individual wells were exposed to the appropriate light conditions through the custom-made light-addressable multiwell plate (*Appendix 1—figure 3*). The red and green light intensities were respectively: 9.82 W/m², 5.64 W/m². Data were analysed through custom python script. Blank absorbance of the medium was removed. Absorbance was then used to normalize fluorescence with respect to cell count.

## Plate reader experiments to detect constructs expression range

Samples were grown from glycerol stocks overnight as in the previous protocol. Cells were refreshed in imaging medium plus antibiotics and OD600 was set to 0.002. Samples were transferred to a 96-well plate and placed in a shaking incubator at 37 °C. There, individual wells were exposed to homogeneous levels of red light and a gradient of green light with the custom device in *Appendix 1—figure 3*. Absorbance and sfGFP fluorescence were measured every hour by temporarily moving the plate from the incubator to the plate reader. Data were analysed through custom python script. Protein production rate was calculated as the time derivative of sfGFP concentration divided by OD600 in a selected OD600 interval of exponential growth.

## Mother machine fabrication

Master molds with mother machine features were fabricated using a hybrid technique involving standard soft-lithography and two-photon polymerization (*Vanderpoorten et al., 2019*). Firstly, the feeding channel (50 μm-width, 15 μm-height) was fabricated using standard protocols of soft-lithography. Precisely, a SU-8 layer of 15 μm was fabricated by spinning SU-8 2015 onto a soda-lime coverglass (3000 rpm - 30 s, Laurell WS-650Mz-23NPPB). Strong adhesion of the SU8 to the carrier coverglass was ensured by three layers of OmniCoat adhesion promoter (MicroChem Corp). The coverglass was soft-baked, and a microfluidic mask with a pattern of 50 μm-wide channels was projected onto the photoresist (UV KUB 2, 400 mJ - 40 s at 25% of maximum power). The coverglass was soft-baked, post-baked, and developed according to the protocol provided by MicroChem Corp. In addition, the sample was hard-baked for 30 min at 170 °C. A second layer of 20 μm was fabricated on top of the feeding channel by spinning SU-8 2015 at 2000 rpm, followed by a soft-baking of 2 hours at 95 °C. Then, the microfabrication of the micron-size channels was carried out by a custom-built two-photon polymerization setup (*Vizsnyiczai et al., 2017*). The refractive index contrast between cured and uncured SU-8 was used to identify the edge of the largest channel, where a comb of micrometer-sized channels was made by direct laser writing. The whole coverglass was again baked at 95°C and rinsed with PGMEA and IPA. Finally, the master mold was silanized to prevent the PDMS from adhering to the master. PDMS was prepared from a Sylgard 184 Silicone Elastomer kit: polymer base and curing

agent were mixed in a 10:1 ratio, and air bubbles were removed from the mixture by centrifugation. The degassed mixture was then poured over the master, and the devices were cured for about 1.5 hr at 90 °C. PDMS chips were peeled from the master mold and bonded to glass by oxygen plasma treatment (15 s at 100 watts) and then baked for 10 min at 120°C.

## Microscopy

Phase contrast and epi-fluorescence imaging were performed using a custom-built optical microscope equipped with a $100\times$ magnification objective (Nikon MRH11902; NA=1.3) and a high-sensitivity CMOS camera (Hamamatsu Orca-Flash 4.0 V3; see *Appendix 1—figure 12*). For phase contrast imaging, a deep red LED (Thorlabs M730L4) with low inhibition of the optogenetic expression system was used. Light control was achieved by directing light through a 10:90 beam splitter (R:T, Thorlabs BSN10R) positioned under the objective. Green and red light stimuli were provided by two LEDs (Thorlabs M530L4, Thorlabs M660L4, respectively) coupled through a dichroic mirror (Thorlabs DMLP567R) and passing through a long-pass filter at 550 nm (Thorlabs FELH0550). Epi-fluorescence imaging was conducted using a blue LED (Thorlabs M455L4) and a filter set for the CFP (Chroma 49001-ET-ECFP). To enable imaging in both phase contrast and epi-fluorescence, a motorized filter wheel (Thorlabs FW102C) was used to switch between the fluorescence filter set and a long-pass filter at 700 nm (Thorlabs FWLH0700). To maintain sample focus throughout the entire acquisition process, a custom script was developed. This script scans the sample in height and finally moves to the z-plane where the image of a specific fixed structure in the chip has maximum edge enhancement (calculated with the Sobel function). To avoid drift on the x-y plane, we correlate the acquired image of the fixed structure with one acquired at the beginning of the experiment and compute the shift on the x-y plane. The scanning process was performed using a motorized vertical lift stage (Zaber X-VSR-E) and a motorized microscope stage (Zaber X-ASR100B120B-SE03D12).

## Mother machine experiments

Cells were grown from a glycerol stock for 16 hr in LB with appropriate antibiotics, in a 50 mL centrifuge tube exposed to saturating 660 nm red light. Bacteria were refreshed 1:100 in imaging medium under red light until they reached an OD of 0.1 (around 4 hours). 1 mL of culture was centrifuged in an Eppendorf miniSpin Plus centrifuge at 1000 rcf for 5 min and concentrated 50 x. 20 µL of concentrated bacteria were loaded in the mothermachine's flow channel with a pipette. Bacteria were loaded in the growth channels, by centrifugation of the mothermachine chip for 7 min at 1300 rcf (with soft ramp acceleration) in an Eppendorf Centrifuge 5430 R using the Combislide adapter. The flow channel of the mothermachine was then washed for approximately 5 min with LB with antibiotics and 0.1% BSA at a constant flow of 50 µL/min, to carry away excess bacteria. For the whole duration of the experiments, the mothermachine was perfused with LB with antibiotics and 0.1% BSA at a constant rate of 5.5 µL/min. Green and red light stimuli were provided by the two LEDs (Thorlabs M530L4, Thorlabs M660L4) with respective intensities 6W/m² and 26W/m² for the synchronization experiments, and 1.1W/m² and 4.5W/m² for the entrainment experiments.

## Data analysis: segmentation in the mother machine

Phase contrast images were taken every 3 min and fluorescence images every 9 min. Phase contrast images were analyzed, to identify the single feeding channels of the microfluidic chip. Single cells were segmented using the pretrained neural network model Cellpose 2.0. We further trained the model with our own data to increase the accuracy of segmentation (*Pachitariu and Stringer, 2022*). The masks obtained after segmentation were used to measure the total fluorescence of the mother cell in all feeding channels of the mother machine as a function of time.

## Simulations

All simulations were performed by direct Euler integration of *Equation 2* using custom Python program. The selection of model parameters was motivated as follows: the mean value of the growth rate was directly estimated from experiments $\alpha = 0.75 \mathrm{h}^{-1}$ (*Appendix 1—figure 2*); for all promoters we choose a cooperativity index $n = 3$ to both match experimental oscillation profiles and in agreement with *Potvin-Trottier et al., 2016*; for the production rates we chose the value $\beta = 300$ h⁻¹ to

match the experimental period $T_0 \simeq 17.5$ h; the maximum light induced production rate $\beta' = 80 \text{ h}^{-1}$ was estimated to maximize agreement with experimental data in *Figure 5a*.

## Acknowledgements

The research leading to these results has received funding from the European Research Council under the ERC Grant Agreement No. 834615 (GF, NC, RDL) and from the Italian Ministry of University and Research (MUR) under the FARE2020 Grant R20R4X8ZEL (MCC, RDL).

## Additional information

### Funding

| Funder | Grant reference number | Author |
| --- | --- | --- |
| European Research Council | 834615 | Roberto Di Leonardo |
| Ministero dell'Università e della Ricerca | R20R4X8ZEL | Roberto Di Leonardo |

The funders had no role in study design, data collection and interpretation, or the decision to submit the work for publication.

### Author contributions

Maria Cristina Cannarsa, Data curation, Formal analysis, Validation, Investigation, Visualization, Methodology, Writing – original draft, Writing – review and editing, Design and construction of all plasmids and strains; Filippo Liguori, Data curation, Formal analysis, Validation, Investigation, Methodology, Writing – original draft, Writing – review and editing, Collaboration to plasmid design; Nicola Pellicciotta, Data curation, Formal analysis, Validation, Investigation, Methodology, Writing – original draft; Giacomo Frangipane, Data curation, Formal analysis, Supervision, Validation, Investigation, Methodology, Writing – original draft, Writing – review and editing, Collaboration to plasmid design; Roberto Di Leonardo, Conceptualization, Formal analysis, Supervision, Funding acquisition, Validation, Visualization, Methodology, Writing – original draft, Writing – review and editing

### Author ORCIDs

Maria Cristina Cannarsa https://orcid.org/0009-0009-4126-4022
Filippo Liguori https://orcid.org/0000-0003-2859-6953
Nicola Pellicciotta https://orcid.org/0000-0002-7105-9566
Giacomo Frangipane https://orcid.org/0000-0002-1533-4754
Roberto Di Leonardo https://orcid.org/0000-0002-5020-0663

Reviewer #1 (Public review): https://doi.org/10.7554/eLife.97754.3.sa1
Reviewer #2 (Public review): https://doi.org/10.7554/eLife.97754.3.sa2
Author response https://doi.org/10.7554/eLife.97754.3.sa3

## Additional files

### Supplementary files

• MDAR checklist

• Source data 1. Plate reader data of optorepressilator detuning (*Figure 6b*, *Figure 5d*), optorepressilator light-driven synchronization (*Figure 4a*) and of repressilator desynchronization (*Figure 1b*).

• Source data 2. Plate reader data of GFP protein expression levels (*Figure 3b*).

• Source data 3. Data of mother machine experiments (*Figure 4b*, *Figure 5e*).

• Source data 4. Plate reader data of the comparison between optorepressilator versions

(*Figure 3d*).

### Data availability

All data generated or analysed during this study are included in the manuscript and supporting files.

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

## Appendix 1

### 1. Analytical solution of repressilator dynamics in the digital approximation

In the digital approximation, Hill functions in *Equation A1* in the main text are replaced by $\Theta$ functions and equations become piecewise linear first order. Oscillations are therefore built by joining together rising (to steady value $\beta/\alpha$) or decaying (to 0) exponentials with rate $\alpha$, calling with $M$ and $m$ respectively the maximum and minimum values in the oscillations. $t$ is the interval between two maxima of two different proteins and $\delta$ is the time interval to rise from the minimum $m$ to 1 (*Appendix 1—figure 1*). All concentrations are measured in units of $K$ so 1 represents the activation threshold. We can write the following four equations in the unknowns $m, M, t, \delta$:

$$Me^{-(t-\delta)} = 1 \tag{A1}$$

$$e^{-(t+\delta)} = m \tag{A2}$$

$$\beta + (1-\beta)e^{-t} = M \tag{A3}$$

$$\beta + (m-\beta)e^{-\delta} = M \tag{A4}$$

Where, for simplicity of notation, we measure time in units of $\alpha^{-1}$.

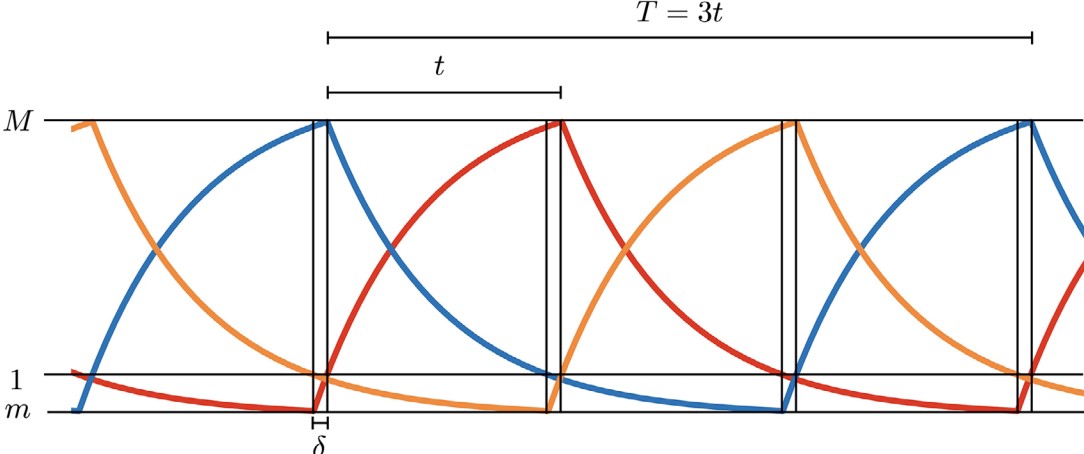

**Appendix 1—figure 1.** Simulations of the digital model. $M$ and $m$, respectively, are the maximum and minimum values in the oscillations and is the time interval to rise from the minimum $m$ to 1 (threshold value in $K$ units).

From *Equation A1 and A2* we find:

$$e^{-2t} = m/M \tag{A5}$$

and

$$e^{-2\delta} = Mm \tag{A6}$$

which substituted in (*Equation A3*) and (*Equation A4*) give:

$$\beta + (1-\beta)\sqrt{m/M} = M \tag{A7}$$

$$\beta + (m-\beta)\sqrt{Mm} = 1 \tag{A8}$$

The two equations above can be solved numerically to get $M$ and $m$ once $\beta$ is known. However if, as in our case, $\beta \gg 1$ we can find the approximate solutions:

$$M = \beta, \quad m = 1/\beta \tag{A9}$$

Substituting back in (*Equation A5*) we get

$$t = \log \beta \tag{A10}$$

and thus for the entire period

$$T = 3t = 3 \log \beta \tag{A11}$$

Finally reintroducing the time scale $\alpha$ we can write for the period

$$T = \frac{3}{\alpha} \log \frac{\beta}{\alpha} \tag{A12}$$

## 2. Growth rate estimate

Growth rate analysis of the optorepressilator data displayed in *Figure 6b* in the main text.

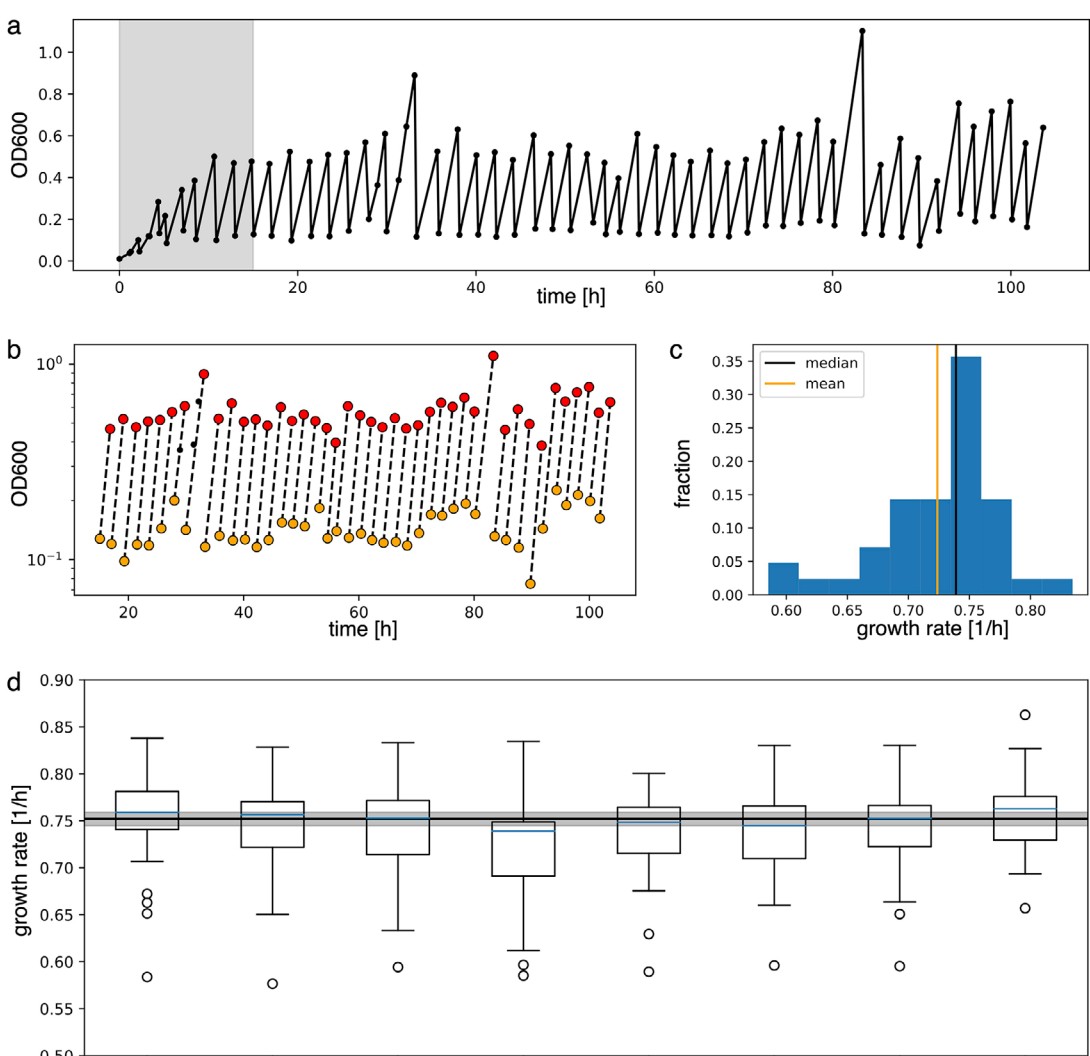

**Appendix 1—figure 2.** Analysis to estimate growth rate from experimental data. (**a**) Raw absorbance data from one individual sample in the plate reader experiment. The initial gray area is removed from further analysis to ensure samples reached stable growth conditions. (**b**) Minimum and maximum OD600 points (respectively in yellow and red) in log scale are fitted with a line to extract the growth rate as its slope. (**c**) Distribution of the growth rates. Median and mean of the distribution are shown respectively in black and orange. Due to the non-symmetric distribution, we used the median to summarize the growth rate. (**d**) Box plots of the growth rates for each sample, all grown in parallel. The blue line inside the box plots is the median. The black line is the mean of all medians, our estimate growth rate ($0.75 \pm 0.01 h^{-1}$) for the optorepressilator in plate reader experiments. The gray area represents the mean growth rate $\pm$ the standard deviation.

## 3. Custom-made light-addressable multiwell plate

The custom-made light-addressable multiwell plate was made using an Adafruit NeoPixel NeoMatrix 8×8, a matrix of 64 RGB LEDs, attached to the bottom of a Greiner 96 well plate with black wells and removed bottoms (*Appendix 1—figure 3a*). The matrix was placed so that each LED was directly under an individual well. An opaque foam layer was cut in order to accommodate LEDs, adjusting the matrix to the plate and reducing well-to-well light contamination. An additional plate was fixed on top of this structure to decrease light intensity on samples by increasing the distance between LEDs and samples. Opaque foam layers were cut in correspondence with the wells and attached above and below this plate, again to avoid well-to-well light contamination. The construct was fixed to a shaking incubator (*Appendix 1—figure 3b*). A 96 well plate could be placed on top of it and removed for data acquisition. A darkened lid was placed on the samples' plate to block environment light. The LEDs matrix was controlled with an Arduino microcontroller. To ensure correct timing of inputs, we employed an AZDelivery Real Time Clock module. LEDs intensity on samples was measured with Thorlabs Standard Photodiode Power Sensor S121C.

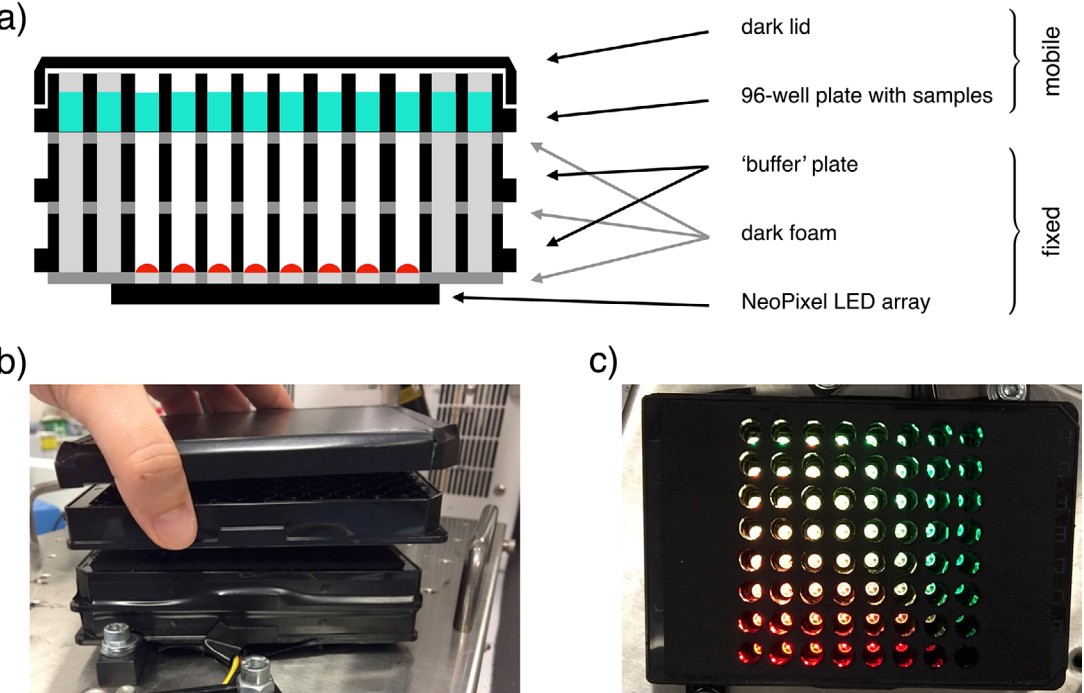

**Appendix 1—figure 3.** Illustration of the light-addressable multiwell plate. (**a**) Scheme of a longitudinal section of the device, with key components highlighted. (**b**) Frontal picture of the custom-made light-addressable multiwell plate placed in a shaking incubator. The sample plate and dark lid are partially lifted. (**c**) Top-down picture of the device. Dark lid and sample plate are removed and LEDs are on.

## 4. Period normalized by growth rate

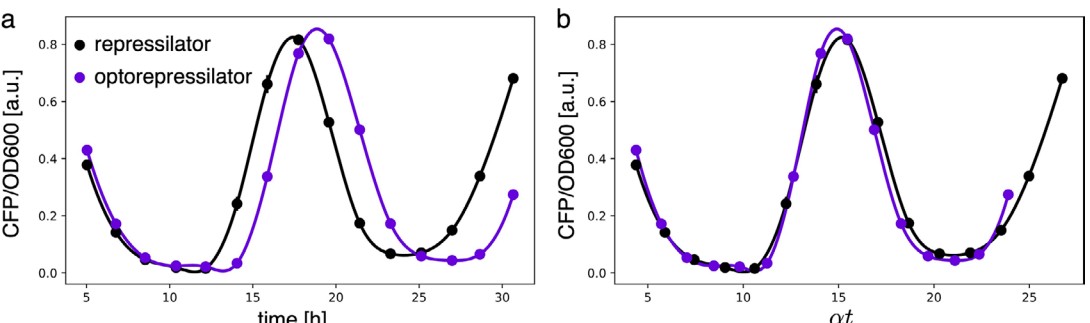

**Appendix 1—figure 4.** Period of different constructs overlaps when normalized by growth rate. (**a**) Fluorescence intensities of the repressilator and optorepressilator, as shown in *Figure 3d*. (**b**) Time axis rescaled: time points of samples in a multiplied by their respective growth rates $\alpha$ (0.87 h$^{-1}$, 0.76 h$^{-1}$) and aligned to a common initial time. Post-rescaling, the oscillations of the repressilator and optorepressilator overlap nicely.

## 5. Effect of leakage on the period and stability of the oscillations

Leakage from the optogenetic $x'$ promoter can destroy limit cycle oscillations and collapse the system into a stable fixed point. To study how leakage affects oscillations we employ linear stability analysis. In the absence of leakage ($\beta' = 0$), the system of equations is symmetric and the fixed point is ($\xi_0, \xi_0, \xi_0$), where $\xi_0$ satisfies the following condition:

$$\xi_0 = \frac{\beta/\alpha}{1 + \xi_0^n}. \tag{A13}$$

Leakage $\beta'$ makes the system asymmetric and the new fixed point coordinates ($x_0, y_0, z_0$) satisfy the following conditions:

$$x_0 = \frac{\beta/\alpha}{1 + z_0^n} + \beta'/\alpha \qquad y_0 = \frac{\beta/\alpha}{1 + x_0^n} \qquad z_0 = \frac{\beta/\alpha}{1 + y_0^n} \tag{A14}$$

Linearizing around the fixed point, we can write for the displacements:

$$\dot{\delta x} = -\gamma_x \delta z - \alpha \delta x \quad \dot{\delta y} = -\gamma_y \delta x - \alpha \delta y \quad \dot{\delta z} = -\gamma_z \delta y - \alpha \delta z \tag{A15}$$

with:

$$\gamma_x = \beta \frac{n z_0^{n-1}}{(1 + z_0)^2} \qquad \gamma_y = \beta \frac{n x_0^{n-1}}{(1 + x_0)^2} \qquad \gamma_z = \beta \frac{n y_0^{n-1}}{(1 + y_0)^2} \tag{A16}$$

Rewriting *Equation A15* in matrix form we have:

$$\Delta = -\Gamma \cdot \Delta \text{ where } \Delta = \begin{pmatrix} \delta_x \\ \delta_y \\ \delta_z \end{pmatrix} \quad \Gamma = \begin{pmatrix} \alpha & 0 & \gamma_x \\ \gamma_y & \alpha & 0 \\ 0 & \gamma_z & \alpha \end{pmatrix} \tag{A17}$$

The three eigenvalues of $\Gamma$ are $\alpha - \bar{\gamma}e^{i\pi/3}, \alpha + \bar{\gamma}, \alpha - \bar{\gamma}e^{-i\pi/3}$, where $\bar{\gamma} = \sqrt[3]{\gamma_x \gamma_y \gamma_z}$. The fixed point is stable when all the eigenvalues have a positive real part ($\bar{\gamma}/\alpha < 2$). At $\bar{\gamma}/\alpha = 2$ we have an Hopf bifurcation and for $\bar{\gamma}/\alpha > 2$ two eigenvalues acquire a negative real part and a limit cycle arises. We numerically calculated $\bar{\gamma}/\alpha$ as a function of normalized leakage $\beta'/\alpha$ for the same parameters we used in all the simulations ($\beta = 300$ h$^{-1}$, $\alpha = 0.75$ h$^{-1}$, $n = 3$). Results are shown in *Appendix 1—figure 5* showing that the Hopf bifurcation occurs for $\beta'/\alpha = 3.24$.

When no leakage is present, $\beta'$ represents the production of $x'$ induced by green light. As discussed in the main text we estimate an experimental value for $\beta'/\alpha = 107$. This means that in all

experiments reported here, light intensity is always strong enough to break limit cycle oscillations if continuously exposed.

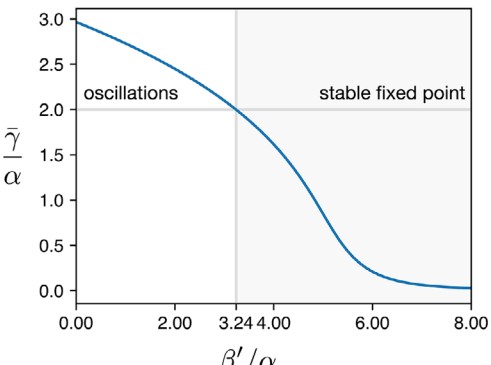
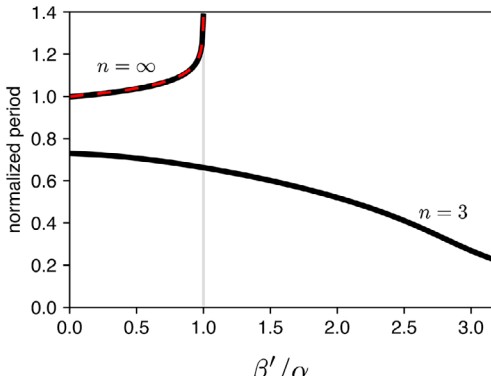

**Appendix 1—figure 5.** Simulations of leakage's effect on limit cycle and period. (**a**) $\bar{\gamma}$ vs leakage shows that increasing the leakage past a certain threshold the existing limit cycle breaks down and the system does damped oscillations towards the fixed point. (**b**) The period of the oscillator is affected by the leakage. We plot the curves for the two hill coefficients treated in the manuscript. For $n = 3$ the period is monotonically decreasing in response to leakage increase, suggesting that the period increase in the optorepressilator respect to the repressilator (**Figure 3d** main text) is not leakage. In the digital case ($n = \infty$) we see that the leakage breaks the limit cycle for values of $\beta'/\alpha \geq 1$.

## 6. Quantification of gene expression levels as a function of light intensity

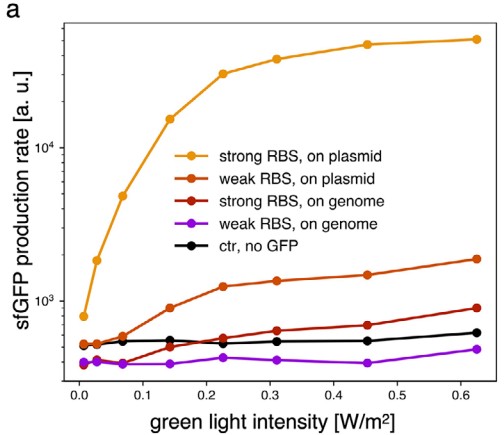

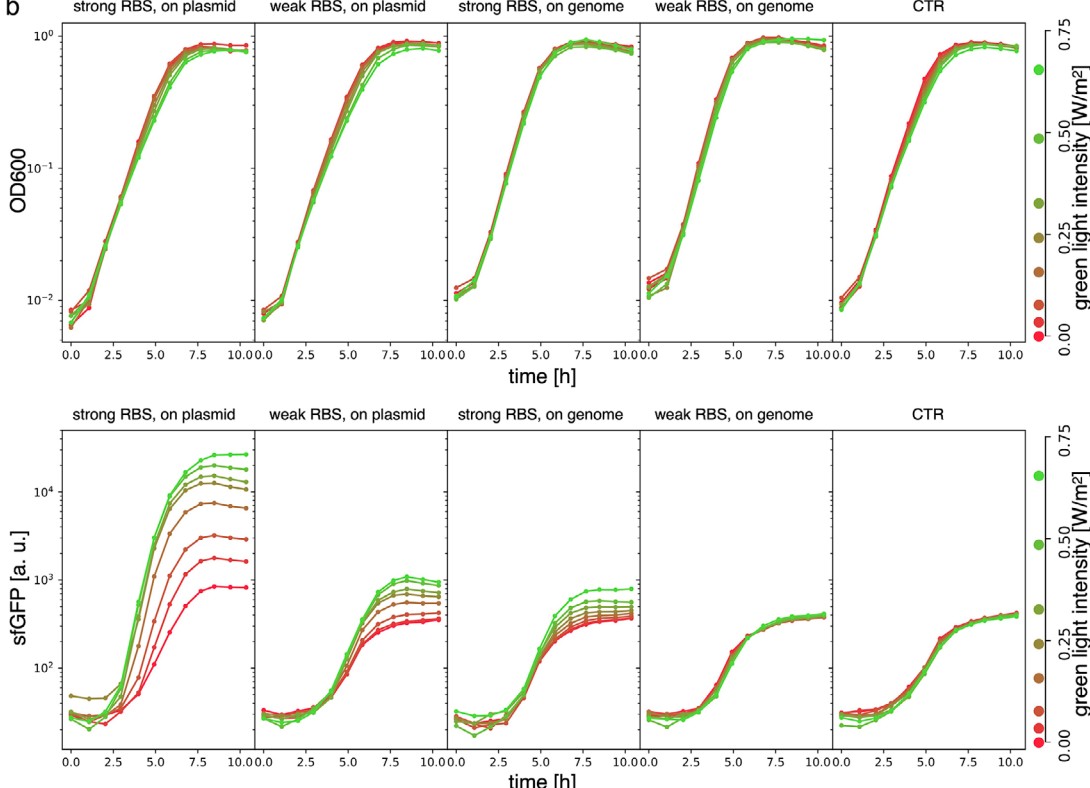

**Appendix 1—figure 6.** Samples are DHL708 *E. coli* strain with pNO286-3 plasmid, added with sfGFP light-driven production construct as shown in *Figure 3* of the main text, while control ('ctr') is DHL708+pNO286-3+pSR58-0. (**a**) Protein production rate over green light intensity (same data as *Figure 3b*). While being exposed to different green light intensities, all samples were also constantly exposed to 0.74 W/m² red light. Dots are the mean values of experiments repeated on 3 different days. (**b**) Raw OD600 (top) and sfGFP fluorescence (bottom) data for one of the sfGFP production replicates.

## 7. Dependence of Phase Response Curves (PRC) on $\beta'$ and $\tau$

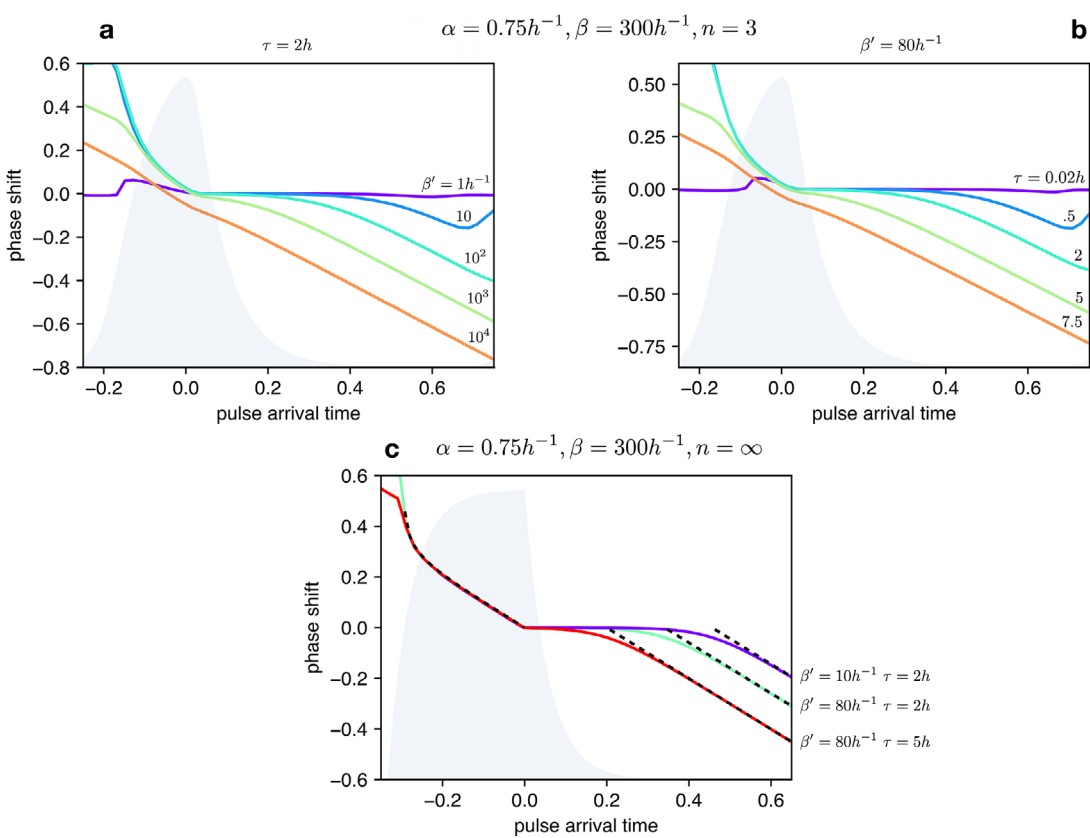

**Appendix 1—figure 7.** Impact of light dependent production rate β′ and input duration $\tau$ on Phase Response Curves. (**a**) PRCs of the optorepressilator for $\tau = 2$ h and $\beta'$ varying logarithmically between 1 h⁻¹ and 10⁴ h⁻¹. (**b**) PRCs of the optorepressilator for $\beta' = 80$ h⁻¹ and $\tau$ varying between 0.02 h and 75 h. (**c**) PRCs of the optorepressilator in the digital approximation ($n = \infty$) for varying $\beta'$ and $\tau$. The black dashed lines represent the theoretical prediction as described below. The other parameters used in the simulations are $\alpha = 0.75$ h⁻¹, $\beta = 300$ h⁻¹, $n = 3$.

Within the digital approximation ($n \to \infty$) we can derive approximate expressions for the phase response curve as follows.

## Case $\phi > 0$

When the pulse arrives at a time delay $t$ after the peak, a bump in $x'$ is produced whose rise time is $\tau$ while the time $t'$ to decay to the threshold value 1 is given by:

$$\frac{\beta'}{\alpha}\left(1 - e^{-\alpha\tau}\right)e^{-\alpha t'} = 1$$

If $x'$ decays to the threshold after $x$ than the next $y$ peak will be delayed by a time:

$$t + \tau + t' + \frac{T_0}{3} - T_0$$

corresponding to a negative phase shift:

$$\Delta\phi = -\left(\phi + \frac{1}{\alpha T_0}\log\left[\frac{\beta'}{\alpha}\left(e^{\alpha\tau} - 1\right)\right] - \frac{2}{3}\right)$$

The predicted PRC as a negative slope equal to −1 and a intercept that depends on the intensity and duration of the light pulse.

## Case $\phi < 0$

When the pulse arrives before the peak ($t < 0$) it triggers the decay of $y$ (blue trace) which will reach the threshold 1 after a time $t'$ given by:

$$\frac{\beta}{\alpha}\left(1 - e^{-\alpha(T_0/3+t)}\right)e^{-\alpha t'} = 1$$

When $y$ reaches the threshold, $z$ production will be turned on and quickly reach the threshold for repressing $x$. From this moment $x$ will start decaying and reach the threshold after a time $t''$:

$$\frac{\beta}{\alpha}\left(1 - e^{-\alpha(t'+t)}\right)e^{-\alpha t''} = 1$$

Using the previously obtained expression for the period $T_0 = 3\log(\beta/\alpha)/\alpha$ we can finally find that the phase will be positively shifted by:

$$\Delta\phi = \frac{1}{\alpha T_0}\log\left[\frac{\beta}{\alpha}e^{\alpha T_0\phi} - 2\right] - \frac{1}{3}$$

This time the PRC is insensitive to the intensity and duration of the light pulse.

These two analytic expressions for the PRC are reported as black dashed lines in *Appendix 1—figure 7*.

## 8. Exploring "Arnold surface" at low forcing

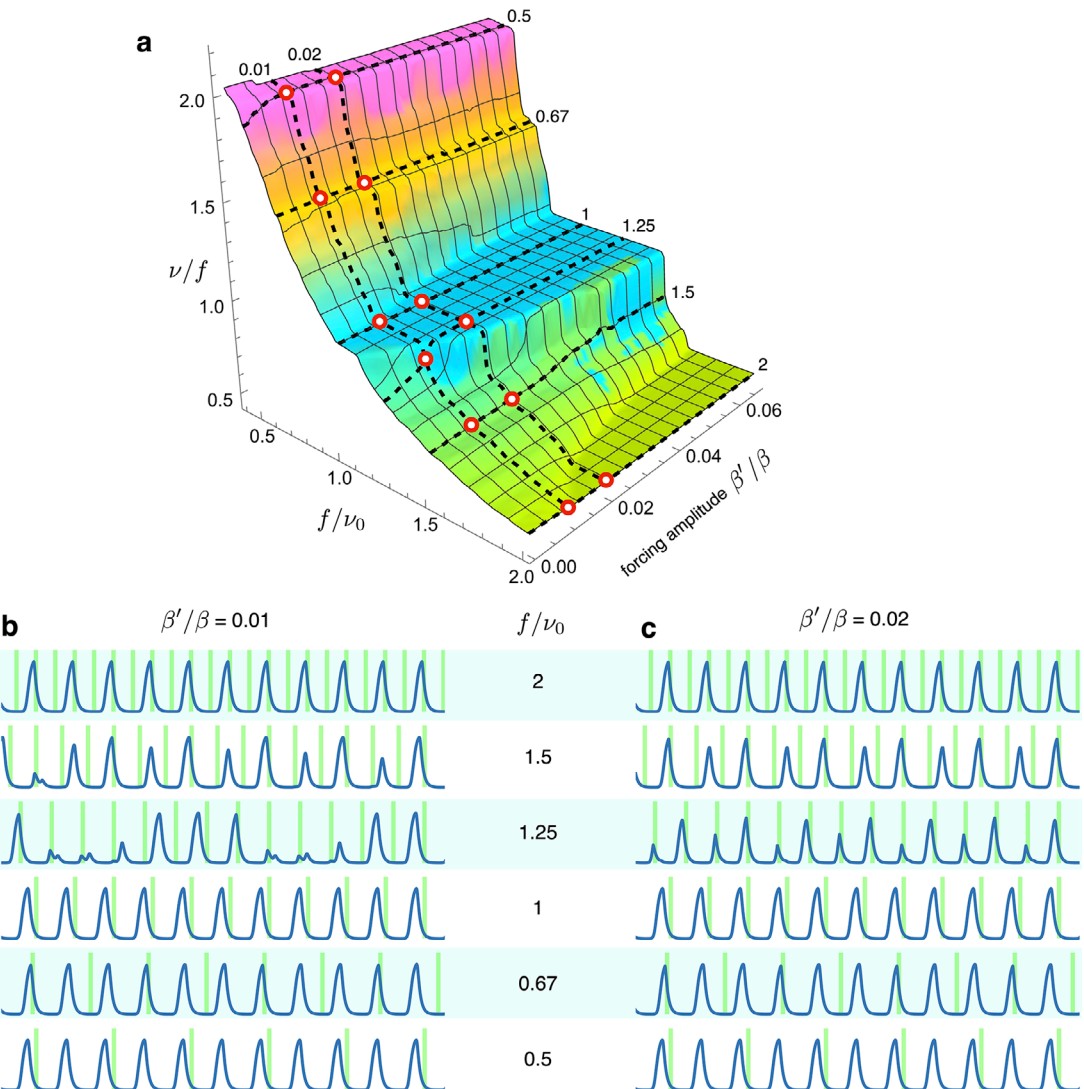

**Appendix 1—figure 8.** Simulations of optorepressilators' response to low forcing amplitude at various frequencies. (**a**) Arnold surface calculated as in *Figure 6* of main text, with red circles to highlight the values of forcing amplitudes and frequencies used for plots in (**b** and **c**). We notice that at a forcing frequency ratios of 1.5 and 1.25, entrainment is achieved only for the higher amplitude values. Interestingly, when moving in between the main plateaus in the Arnold surface we find entrainment at fractional frequency ratios such as $2/3\,f$ at $f/\nu_0 = 1.5$ and $3/2\,f$ at $f/\nu_0 = 0.67$.

## 9. Correlation of period and amplitude with growth rate

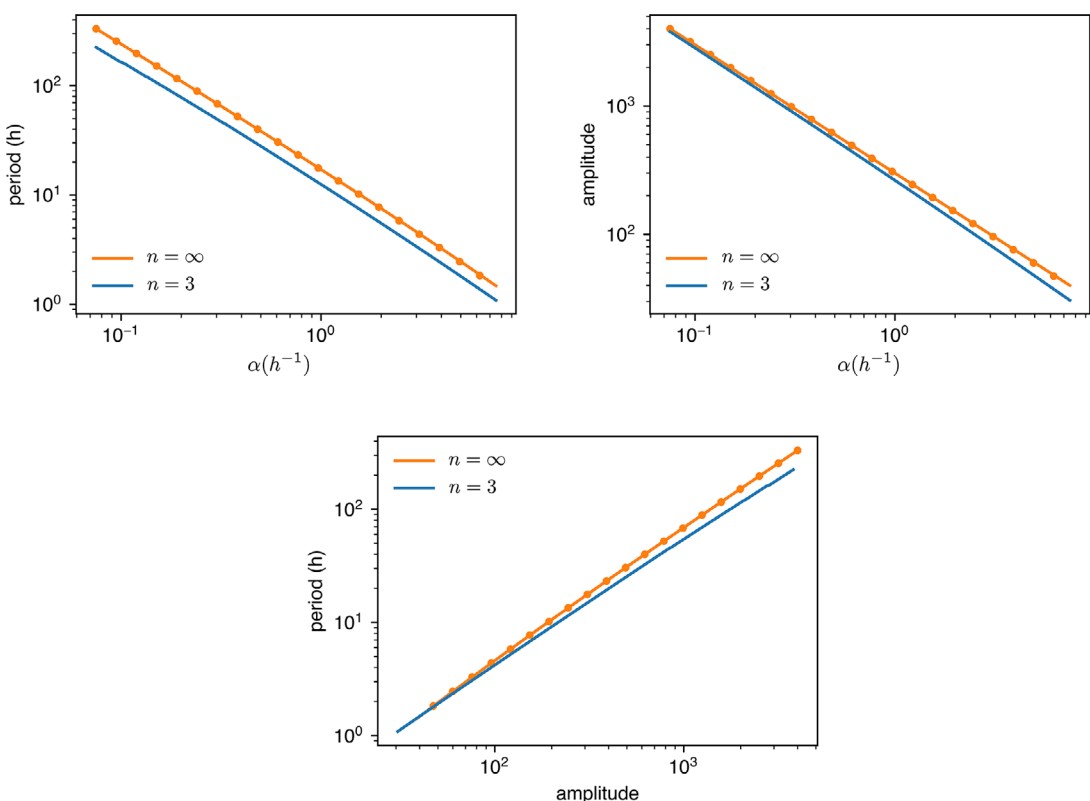

**Appendix 1—figure 9.** $\alpha$ values are changed from $1/10$ of the standard value $(0.75\text{h}^{-1})$ to 10 times as much. Amplitude is dimensionless, in units of $K$. Dots are simulations, the continuous lines are predictions with $n = 3$ or digital approximation $n = \infty$.

## 10. Strains

**Appendix 1—table 1.** *Escherichia coli* strains used in this work.

| Bacterial strain | Genotype | Source | Parent |
|---|---|---|---|
| DHL708 | MC4100$\Delta clpPX$ | *Potvin-Trottier et al., 2016* | |
| MCC0234 | DHL708 $\Delta attB :: lacI34$ $P_{cpcG2-172}$ $BBa\_B0034$ $lacI$ $(kan)$ | this work | DHL708 |
| MCC0233 | DHL708 $\Delta attB :: lacI33P_{cpcG2-172}$ $BBa\_B0033$ $lacI$ $(kan)$ | this work | DHL708 |
| MCC0034 | DHL708 $\Delta attB :: gfp34P_{cpcG2-172}$ $BBa\_B0034$ $sfgfp$ $(kan)$ | this work | DHL708 |
| MCC0033 | DHL708 $\Delta attB :: gfp33P_{cpcG2-172}$ $BBa\_B0033$ $sfgfp$ $(kan)$ | this work | DHL708 |

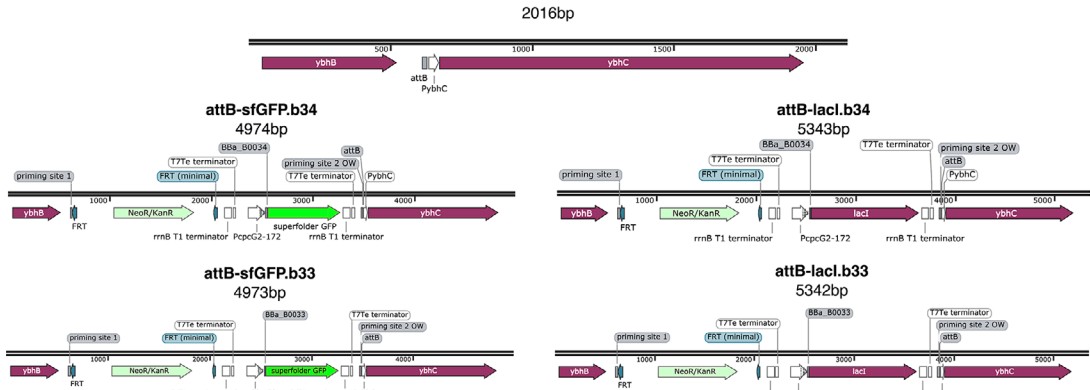

**Appendix 1—figure 10.** Maps of the original *attB* site and genome insertions. Genome sites correspond to *E. coli* strains: MCC0034 top left, MCC0033 bottom left, MCC0234 top right, MCC0233 bottom right. All maps are created with SnapGene.

# 11. Plasmids

**Appendix 1—table 2.** Plasmids.

| Plasmid | Description | Source | Parent |
|---|---|---|---|
| pLPT234 | core repressilator componentsand reporters | *Riglar et al., 2019* | |
| pLPT145 | repressors' sponge | *Potvin-Trottier et al., 2016* | |
| pNO286-3 | light sensor andPCB enzymes | *Ong and Tabor, 2018* | |
| pSR58.6 | CcaR, light-drivensfGFP production(RBS BBa_B0034) | *Schmidl et al., 2014* | |
| pSR58.b33 | CcaR, light-drivensfGFP production(RBS BBa_B0033) | this work | pSR58.6 |
| pSR58-0 | CcaR, no $P_{cpcG2-172}$ nor sfGFP | this work | pSR58.6 |
| pSpongeROG.b34 | light-driven LacI production (RBS BBa_B0034), repressors's sponge, CcaR | this work | pSR58.6 (backbone), pLPT145 (sponge), pLPT234 (lacI) |
| pSpongeROG.b33 | light-driven LacIproduction(RBS BBa_B0033),repressors's sponge,CcaR | this work | pSR58.6 (backbone), pLPT145 (sponge), pLPT234 (lacI) |
| pSpongeROG0 | repressors's sponge,CcaR | this work | pSR58.6 (backbone), pLPT145 (sponge) |
| pKD46-sfGFP.b34 | genome insertionof light-drivensfGFP cassette | this work | pKD46 (backbone),pSR58.6 |
| pKD46-sfGFP.b33 | genome insertionof light-drivensfGFP cassette | this work | pKD46 (backbone),pSR58.b33 |
| pKD46-lacI.b34 | genome insertionof light-drivenlacI cassette | this work | pKD46 (backbone),pSpongeROG.b34 |
| pKD46-lacI.b33 | genome insertionof light-drivenlacI cassette | this work | pKD46 (backbone),pSpongeROG.b33 |

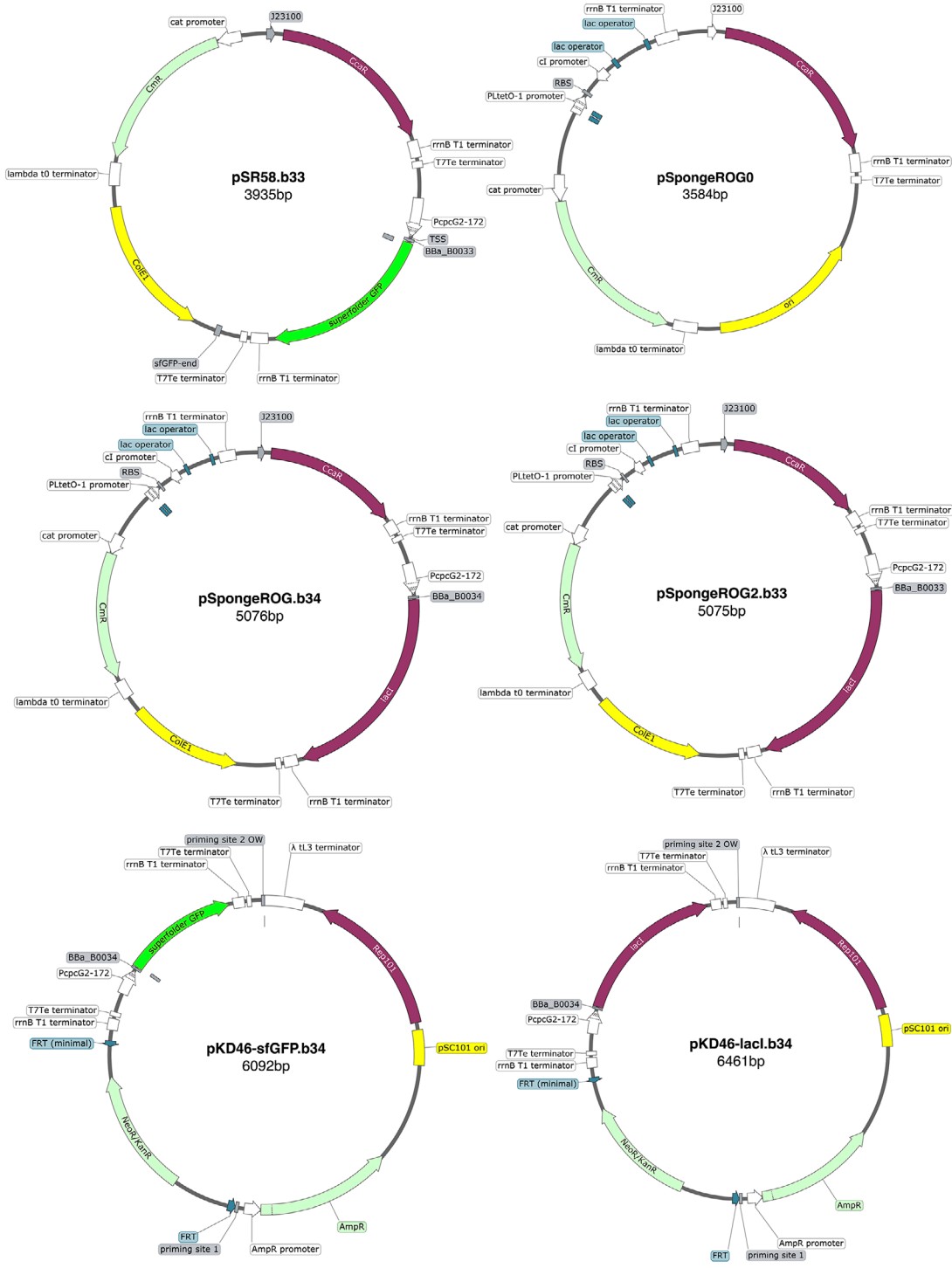

**Appendix 1—figure 11.** Plasmid maps. All maps are created with SnapGene.

## 12. Antibiotic concentrations

**Appendix 1—table 3.** Antibiotic concentrations.

| Antibiotic | Working concentration |
| --- | --- |
| Streptomycin | 50 µg/ml |

*Appendix 1—table 3 Continued on next page*

*Appendix 1—table 3 Continued*

| Antibiotic | Working concentration |
| --- | --- |
| Ampicillin | 100 µg/ml |
| Kanamycin | 50 µg/ml |
| Chloramphenicol | 20 µg/ml |
| Spectinomycin | 100 µg/ml |

## 13. Microscope setup

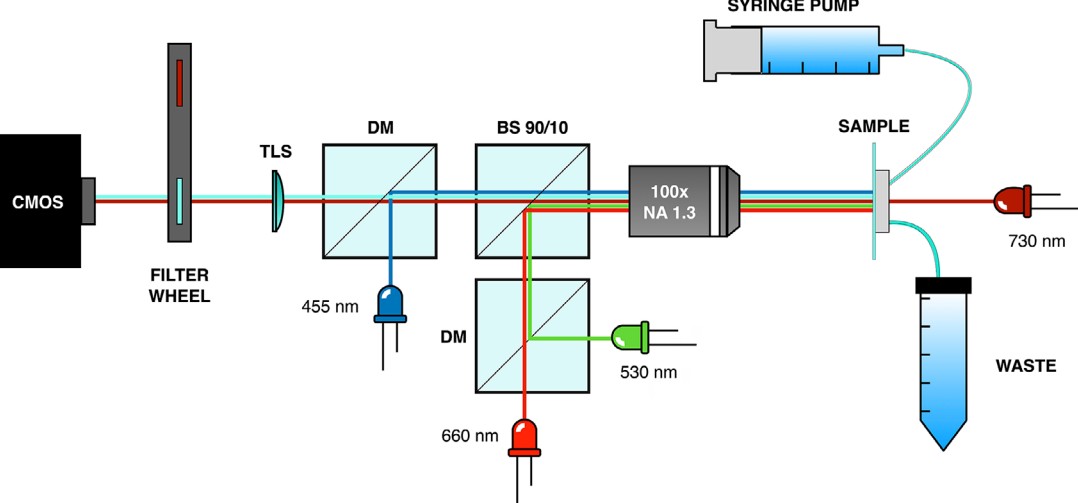

**Appendix 1—figure 12.** Microscope setup used for mothermachine experiments.

