## [Editor Report · eLife assessment]

This study presents a light-entrainable synthetic oscillator in bacteria, the optorepressilator. The authors develop a toolbox using optogenetics that makes the cellular oscillator easily controllable. This toolbox is **valuable**, contributing both to bioengineering and to the understanding of biological dynamical systems. The comparison with a mathematical model, population, and single-cell measurements demonstrate **convincingly** that the planned system was achieved and is suitable to control and study biological oscillators.

---

## [Referee Report · Reviewer #1 (Public review)]

Summary:

The "optorepressilator", an optically controllable genetic oscillator based on the famous *E. coli* 3-repressor (LacI, TetR, CI) oscillator "repressilator", was developed. An individual repressilator shows a stable oscillation of the protein levels with a relatively long period that extends a few doubling times of *E. coli*, but when many cells oscillate, their phases tend to desynchronize. The authors introduced an additional optically controllable promoter through a conformal change of CcaS protein and let it control how much additional CI is produced. By tightly controlling the leak from the added promoter, the authors successfully kept the original repressilator oscillation when the added promoter was not activated. In contrast, the oscillation was stopped by expressing the additional CI. Using this system, the authors showed that it is possible to synchronise the phase of the oscillation, especially when the activation happens as a short pulse at the right phase of the repressilator oscillation. The authors further show that, by changing the frequency of the short pulses, the repressilator was entrained to various ratios to the pulse period, and the author could reconstruct the so-called "Arnold tongues", the signature of entrainment of the nonlinear oscillator to externally added periodic perturbation. The behaviour is consistent with the simplified mathematical model that simulates the protein concentration using ordinary differential equations.

Strengths:

Optical control of the oscillation of the protein clock is a powerful and clean tool for studying the synthetic oscillator's response to perturbation in a well-controlled and tunable manner. The article utilizes the plate reader setup for population average measurements and the mother machine setup for single-cell measurements, and they complement nicely to acquire necessary information.

---

## [Referee Report · Reviewer #2 (Public review)]

Summary:

In this manuscript by Cannarsa et. al., the authors describe the engineering of a light-entrainable synthetic biological oscillator in bacteria. It is based on an upgraded version of one of the first synthetic circuits to be constructed, the repressilator. The authors sought to make this oscillator entrainable by an external forcing signal, analogous to the way natural biological oscillators (like the circadian clock) are synchronized. They reasoned that an optogenetic system would provide a convenient and flexible means of manipulation. To this end, the authors exploited the CcaS-CcaA light-switchable system, which allows activation and deactivation of transcription by green and red light, respectively. They used this system to make the expression of one of the repressilator's transcription factors (lacI) light-controlled, from a construct separated from the main repressilator plasmid. This way, under red light the oscillator runs freely, but exposure to green light causes overexpression of the lacI, pushing the system into a specific state. Consequently, returning to red light will restore the oscillations from the same phase in all cells, effectively synchronizing the cell population.

After demonstrating the functionality of the basic concept, the authors combined modeling and experiments to show how periodic exposure to green light enables efficient entrainment, and how the frequency of the forcing signal affects the oscillatory behavior (detuning).

This work provides an important demonstration of engineering tunability into a foundational genetic circuit, expands the synthetic biology toolbox, and provides a platform to address critical questions about synchronization in biological oscillators. Due to the flexibility of the experimental system, it is also expected to provide a fertile ground for future testing of theoretical predictions regarding non-linear oscillators.

Strengths:

* The study provides a simple and elegant mechanism for the entertainment of a synthetic oscillator. The design relies on optogenetic proteins, which enable efficient experimentation compared to alternative approaches (like using chemical inducers). This way, a static culture (without microfluidics or change of growth media) can be easily exposed to flexible temporal sequences of the zeitgeber, and continuously measured through time.

* The study makes use of both plate-reader-based population-level readout and mother-machine single-cell measurements. Synchronization through entrainment is a single cell level phenomenon, but with a clear population-level manifestation. Thus, this experimental approach combination provides a strong validation to their system. At the same time, differences between the readout from the two systems have emerged, and provided a further opportunity for model refinement and testing.

* The authors correctly identified the main optimization goal, namely the effective leakiness of their construct even under red light. Then, they successfully overcame this issue using synthetic biology approaches.

* The work is supported by a simplified model of the repressilator, which provides a convenient analytical and numerical means to draw testable predictions. The model predictions are well aligned with the experimental evidence.

Weaknesses:

* Even after optimizing the expression level of the light-sensitive gene, the system is very sensitive, i.e., a very short exposure is sufficient to elicit the strongest entertainment. This limited dynamic range might hamper some model testing and future usage.

* As a result of the previous point, the system is entrained by transiently "breaking" the oscillator: each pulse of green light represents a Hopf bifurcation into a single attractor. it means that the system cannot oscillate in constant green light. In comparison, this is generally not the case for natural zeitgebers like light and temperature for the circadian rhythms. Extreme values might prevent oscillations (not necessarily due to breaking the core oscillator), but usually, free running is possible in a wide range of constant conditions. In some cases, the free-running period length will vary as a function of the constant value.

While the approach presented in this manuscript is valid, a comprehensive analysis of more subtle modes of repressilator entrainment could also be of value.

* The entire work makes use of a single intensity and single duration of the green pulse to force entrainment. While the model has clear predictions for how different modalities should affect entrainment, more experiments are needed to validate those predictions.

---

## [Author Response]

The following is the authors’ response to the original reviews.

**Public Reviews.**

**Reviewer #1 (Public Reviews):**
Summary:The "optorepressilator", an optically controllable genetic oscillator based on the famous *E. coli* 3-repressor (LacI, TetR, CI) oscillator "repressilator", was developed. An individual repressilator shows a stable oscillation of the protein levels with a relatively long period that extends a few doubling times of *E. coli*, but when many cells oscillate, their phases tend to desynchronize. The authors introduced an additional optically controllable promoter through a conformal change of CcaS protein and let it control how much additional CI is produced. By tightly controlling the leak from the added promoter, the authors successfully kept the original repressilator oscillation when the added promoter was not activated. In contrast, the oscillation was stopped by expressing the additional CI. Using this system, the authors showed that it is possible to synchronise the phase of the oscillation, especially when the activation happens as a short pulse at the right phase of the repressilator oscillation. The authors further show that, by changing the frequency of the short pulses, the repressilator was entrained to various ratios to the pulse period, and the author could reconstruct the so-called "Arnold tongues", the signature of entrainment of the nonlinear oscillator to externally added periodic perturbation. The behaviour is consistent with the simplified mathematical model that simulates the protein concentration using ordinary differential equations.Strengths:Optical control of the oscillation of the protein clock is a powerful and clean tool for studying the synthetic oscillator's response to perturbation in a well-controlled and tunable manner. The article utilizes the plate reader setup for population average measurements and the mother machine setup for single-cell measurements, and they compensate nicely to acquire necessary information.Weakness:The current paper added the optogenetically controlled perturbation to control the phase of oscillation and entrainment, but there are a few other works that add external perturbation to a collection of cells that individually oscillate to study phase shift and/or entrainment. The current paper lacks discussion about the pros and cons of the current system compared to previously analyzed systems.RecommendationsEven if the main purpose of the current paper is to develop a toolbox, it is beneficial to emphasize the pros and cons of the current system compared to the existing work. In addition to the ref [36] that authors cite but do not discuss concretely, example literature about entrainment includes:- Sanchez, P.G.L., Mochulska, V., Denis, C.M., Mönke, G., Tomita, T., Tsuchida-Straeten, N., Petersen, Y., Sonnen, K., François, P. and Aulehla, A., 2022. Arnold tongue entrainment reveals dynamical principles of the embryonic segmentation clock. Elife, 11, p.e79575.- Heltberg M, Kellogg RA, Krishna S, Tay S, Jensen MH. Noise induces hopping between NF-κB entrainment modes. Cell systems. 2016 Dec 21;3(6):532-9.There is surely more literature. It is recommended that a solid discussion be added on the relation between existing works and current work.

We thank the Reviewer for their positive comments on our manuscript. Their main recommendation is to expand literature and discuss how our method compares to previously reported entrainment of genetic oscillators. In summary, we believe that the main advantages of the optorepressilator are the simplicity of the transcriptional network combined with the flexibility of optical control. In the “Discussion” section of the revised manuscript, we now try to highlight this also in connection to the suggested literature.

**Reviewer #2 (Public Reviews):**
SummaryIn this manuscript by Cannarsa et. al., the authors describe the engineering of a light-entrainable synthetic biological oscillator in bacteria. It is based on an upgraded version of one of the first synthetic circuits to be constructed, the repressilator. The authors sought to make this oscillator entrainable by an external forcing signal, analogous to the way natural biological oscillators (like the circadian clock) are synchronized. They reasoned that an optogenetic system would provide a convenient and flexible means of manipulation. To this end, the authors exploited the CcaS-CcaA light-switchable system, which allows activation and deactivation of transcription by green and red light, respectively. They used this system to make the expression of one of the repressilator's transcription factors (lacI) light-controlled, from a construct separated from the main repressilator plasmid. This way, under red light the oscillator runs freely, but exposure to green light causes overexpression of the lacI, pushing the system into a specific state. Consequently, returning to red light will restore the oscillations from the same phase in all cells, effectively synchronizing the cell population.After demonstrating the functionality of the basic concept, the authors combined modeling and experiments to show how periodic exposure to green light enables efficient entrainment, and how the frequency of the forcing signal affects the oscillatory behavior (detuning).This work provides an important demonstration of engineering tunability into a foundational genetic circuit, expands the synthetic biology toolbox, and provides a platform to address critical questions about synchronization in biological oscillators. Due to the flexibility of the experimental system, it is also expected to provide a fertile ground for future testing of theoretical predictions regarding non-linear oscillators.Strengths:The study provides a simple and elegant mechanism for the entertainment of a synthetic oscillator. The design relies on optogenetic proteins, which enable efficient experimentation compared to alternative approaches (like using chemical inducers). This way, a static culture (without microfluidics or change of growth media) can be easily exposed to flexible temporal sequences of the zeitgeber, and continuously measured through time.The study makes use of both plate-reader-based population-level readout and mother-machine single-cell measurements. Synchronization through entrainment is a single cell level phenomenon, but with a clear population-level manifestation. Thus, this experimental approach combination provides a strong validation to their system. At the same time, differences between the readout from the two systems have emerged, and provided a further opportunity for model refinement and testing.The authors correctly identified the main optimization goal, namely the effective leakiness of their construct even under red light. Then, they successfully overcame this issue using synthetic biology approaches.The work is supported by a simplified model of the repressilator, which provides a convenient analytical and numerical means to draw testable predictions. The model predictions are well aligned with the experimental evidence.Weaknesses:Even after optimizing the expression level of the light-sensitive gene, the system is very sensitive, i.e., a very short exposure is sufficient to elicit the strongest entertainment. This limited dynamic range might hamper some model testing and future usage.As a result of the previous point, the system is entrained by transiently "breaking" the oscillator: each pulse of green light represents a Hopf bifurcation into a single attractor. it means that the system cannot oscillate in constant green light. In comparison, this is generally not the case for natural zeitgebers like light and temperature for the circadian rhythms. Extreme values might prevent oscillations (not necessarily due to breaking the core oscillator), but usually, free running is possible in a wide range of constant conditions. In some cases, the free-running period length will vary as a function of the constant value. While the approach presented in this manuscript is valid, a comprehensive analysis of more subtle modes of repressilator entrainment could also be of value.The entire work makes use of a single intensity and single duration of the green pulse to force entrainment. While the model has clear predictions for how those modalities should affect entrainment, none of the experiments attempted to validate those predictions.

While we agree with the Reviewer that all reported experiments were performed with pulses of constant amplitude and duration, we do not see this as a necessary limitation for future studies on the optorepressilator. Using pulse-width modulation, green light intensity could be easily and continuously modulated from zero to a maximum value (as in Fig. 4), exploring a wide range of intermediate intensity levels and therefore of mean LacI production rates from the optogenetic promoter. We do not include additional experiments in the revised manuscript but we have greatly expanded the theoretical discussion on the low amplitude regime, both for a constant illumination (new Supplementary Materials Section 5) and the pulsed case (new Supplementary Fig. 8).

**Recommendations for the Authors:**
(1) The introduction emphasized the utility of entrainment as a means to achieve population-wide synchrony. It is worth mentioning also that it enables synchronization of the internal oscillator with an external zeitgeber, to achieve a specific phase-locking between them. Often, this is the main utility attributed to entrainment, e.g., in circadian clocks.

Following Reviewer’s suggestion we now say in the introduction:

These oscillations maintain a constant phase relation to the external light cue that can act as a zeitgeber.

(2) It is sometimes unclear at first glance which of the figure panels show simulation data and which show experimental data (e.g., Figure 5a,b; Figure 6a,b). More explicitly labeling the panels could help.

We thank the reviewer for pointing this out, we now explicitly label all the panels.

(3) Figure 3b - please add a color bar to indicate the meaning of the red-green scale, and enlarge the markers so their color is more visible. Also, can add additional controls of (i) sfGFP expression without the ccaR, and (ii) the autofluorescent signal from wild type. Please also provide the raw data (not the time derivatives) in a supplementary figure.

A colorbar has been added and markers enlarged.

(i) Unfortunately we do not have a control for GFP expression without ccaR.

(ii) autofluorescence signal from “a negative control consisting of DHL708 with plasmids pNO286-3 and pSR58-0 (optogenetic plasmids without sfGFP cassette)” has been added for comparison to Fig.3b. This modification was actually very helpful in understanding that the sensitivity threshold in our experiments is mainly determined by autofluorescence. OD600 and fluorescence raw data are now provided in Supplementary Fig. 6.

(4) Figure 3d - the claim in the text is that the purple optorepressilator and the wildtype repressilator have identical periods and amplitude. However, it seems from the figure that there is a small difference in the period length. This deviation is not problematic in any way, but I wondered whether it might actually be explained by the model, assuming that there is still a very weak leak from the new construct. In other words, would the model predict a bifurcation diagram in which an increasing x' concentration causes a gradual decrease in amplitude and increase in period, before the loss of rhythmicity? If so, Figure 3d can serve not only as a technical optimization demonstration but also as a nice validation of the model.

We thank the reviewer for raising this interesting point. We now report, in Supplementary Materials Section 5, a theoretical prediction of the period with respect to a constant concentration of x'. For our choice of parameters (adjusted to reproduce the main experimental quantitative features) we find a period that decreases with x'. Leakage would therefore lead to a shorter period, contrary to what is observed experimentally. To explain the longer period observed in the optorepressilator we went back to extract the average growth rates of bacteria in the purple optorepressilator and repressilator curves in Fig.3d. As we now discuss in the main text:

“The slight difference in period can be explained by the presence of additional plasmids in the optorepressilator strain, which results in a lower growth rate (Supplementary Figures 4 and 5). As found in the digital approximation, the repressilator period is mainly controlled by the inverse growth rate (see Figure 1a and Supplementary Figure 9) meaning a lower growth rate results in a longer oscillation period. When we normalize the time with the growth rate the two oscillations overlap nicely (Supplementary Figure 4).”

(5) Supplementary Figure 10 has no reference from the main text. it is unclear what's the difference from Figure 3. In general, many items in the supplementary materials are not referenced from the text. In addition, on many occasions, there is a reference to "supplementary information" without a specific address, which is not so useful to the reader. In any case possible, please be more specific. Also, note that there's inconsistency in referring to the supplemental section as "supplementary materials" vs "supplemental information".

We now explicitly reference all Supplementary Figures in the main text and use consistent reference to Supplementary Materials.

(6) The discussion at the bottom of p.7 ("Optogenetic entrainment") is missing a reference to the duration and intensity of the zeitgeber: In the example from human circadian rhythms it doesn't indicate light intensity; In the modeling of the PRC, both modalities are absent. it is important at least to indicate the parameters used for the simulation and experiments. It would be even better to explore in the model how these modalities affect the PRC and entrainment. And it would be incredible if the authors could show this also experimentally.

We now report the light intensity values for:

- our experiments:

“We first demonstrate this by monitoring the population signal from CFP (reporting TetR or 𝑦 in the model) in multiwell cultures under constant red illumination (9.82 W/m^2) interrupted by green light pulses (5.64 W/m^2) with a duration of 2 h and period 𝑇 = 18 h.”

For mother machine experiments “Green and red light stimuli were provided by the two LEDs (Thorlabs M530L4, Thorlabs M660L4) with respective intensities 6 W/m^2 and 26 W/m^2 for the synchronization experiments, and 1.1 W/m^2 and 4.5 W/m^2 for the entrainment experiments.”

- and simulations:

“In Fig. 5a we report the phase shift produced by a single pulse (with duration tau=2 h and intensity beta’=80 h-1 fixed for all the simulations) as a function of the pulse arrival phase ϕ.”

We also added an additional supplementary figure (Supplementary Fig. 7) that explores how the duration and intensity of the light pulses affect the PRC in the model. An approximate analytic result is also derived for the PRC in the digital approximation that compares very well with simulation, providing physical insight into PRC shape (Supplementary Materials Section 7).

(7) The experimental validation of the PRC can be much more thorough. Notably, an entrainment experiment with repeated pulses does not provide the same level of validation as a proper PRC experiment. This is because many differently shaped PRCs can give rise to the same entrainment pattern, as long as their fixed-point phases are the same.Luckily, there might already be a decent amount of data from the mother machine experiments to fit with the PRC prediction, given the authors have pulsed a non-synchronized population that spans the entire x-axis of the PRC. It is possible that a proper PRC experiment wouldn't be too difficult with the plate reader either, given the throughput of the author's system.

This is a very interesting suggestion but unfortunately, in our mother machine data, the first pulse arrives before the cells have completed a full cycle, so although different cells receive the first pulse at a sufficiently randomized phase, we can’t extract their individual phases at the pulse arrival time.

Indeed it would be possible to design a plate reader experiment for the specific purpose of directly measuring the PRC. However, our current protocol involves continuous manual dilutions, which makes it rather laborious. We are currently working on an automated procedure that will allow us to systematically address this and other interesting suggestions in the future.

An indirect experimental validation of the PRC is however still possible using available data. See added red points in Fig.5a and reply to point 10 below.

(8) The discrepancy between the mother machine and plate reader experiments in Figure 5 is explained by a difference in growth rate variability in the two systems. It is not readily obvious how a difference in variability rather than the mean value of the period length can cause a shifted mean phase. It is only hinted in the text that growth rate has two different effects - on the period as well as the amplitude. I hypothesize that because of this period and amplitude correlation, there is a bias contribution to the sum of trajectories that have resulted in a shifted mean phase. Maybe there is another contribution from the asymmetric waveform of the signal? or from the distribution the alpha is sampled from? A direct discussion on that point will make the results much clearer. If the period-amplitude speculation above is right, please add also a panel that shows it. It will also be helpful to show the predicted PRC for the two parameter regimens.

We thank the reviewer for highlighting this point. In the previous version of the manuscript we omitted the fact that in order to better match experimental signals we chose slightly different values for T_L/T_0 for simulations in Fig. 5d and 5e. We now report the values of all simulation parameters in the revised manuscript. This difference could also contribute to the shift in the mean phase for the two cases. We added this information in the main text.

“The bottom panel in Fig. 5d shows the result of a numerical simulation with the same parameters as in Fig. 1b and the addition of a periodic light stimulation, with period $T_L/T_0 = 1$} [...] For the simulations in the lower panel of Fig.5e, all parameters remained the same as in Fig.5d with the exception of the period of the light pulses (T_L/T_0 = 0.97) and the standard deviation of the growth rate distribution, which was increased from 0.034 h^-1 to 0.071 h^-1 to better reproduce the experimental observations in the mother machine.”

Additionally, we added a supplementary figure (Supplementary Fig. 9) demonstrating the correlation between period and amplitude of the oscillations, for simulations with varying growth rate.

(9) The results from the detuning experiments are really nice, especially the decomposition in high frequency shown in Figure 6c. However, the experiments explore only the very high forcing amplitude conditions. Is there any way to test the weaker forcing regimens, as these are expected to uncover the interesting areas in between the Arnold's tongues? If this is experimentally difficult, it would be interesting to include at least the model prediction.

We thank the reviewer for stimulating us to go in this direction. We have performed simulations to explore model predictions for areas between the Arnold’s tongues. We find onset of entrainment as the amplitude increases and also the existence of intermediate plateaus at fractional frequency ratios. These results are now included in the Supplementary Fig. 8.

(10) Another prediction from the Arnold's tongue would be the relative phase of entrainment in different f/v0 conditions. The text refers to it very briefly, but this is a quantitative prediction that can be demonstrated clearly in a figure - how well do they match? It can be shown, for example, by a plot with f/v0 on the x-axis, the phase difference between the pulse and peak expression on the y-axis, a curve representing the model prediction for that function, and dots (with error bars) representing the calculated values from the experimental data.Generally, when suitable, this kind of direct comparison is more useful to the reader than the way the authors chose to compare simulation and experiments throughout the manuscript.

We thank the reviewer for this very interesting suggestion. We have completely rewritten the discussion on entrainment commenting on how the same PRC (phase shift vs pulse arrival phase) can be interpreted as a T_L/T_0-1 vs phase difference plot. Indeed in the new Fig.5a we plot over the theoretical PRC curve, the values of the relative phase of entrainment for three values of the period of the light pulses (from the data in Fig. 6b). The agreement is remarkably good, providing a further experimental validation of the predicted PRC.

(11) The raw data can be valuable for the community for reanalysis and further hypothesis testing. Hence, it will be very useful to make all of the data (e.g., the fluorescence signal quantification tables from all the experiments) publicly available.

We prepared files with all raw data, to be made available to the community.